# Pause sequences facilitate entry into long-lived paused states by reducing RNA polymerase transcription rates

Ronen Gabizon[1], Antony Lee [2], Hanif Vahedian-Movahed [3,4], Richard H. Ebright [3] & Carlos J. Bustamante [1,5,6]

Transcription by RNA polymerase (RNAP) is interspersed with sequence-dependent pausing. The processes through which paused states are accessed and stabilized occur at spatio-temporal scales beyond the resolution of previous methods, and are poorly understood. Here, we combine high-resolution optical trapping with improved data analysis methods to investigate the formation of paused states at enhanced temporal resolution. We find that pause sites reduce the forward transcription rate of nearly all RNAP molecules, rather than just affecting the subset of molecules that enter long-lived pauses. We propose that the reduced rates at pause sites allow time for the elongation complex to undergo conformational changes required to enter long-lived pauses. We also find that backtracking occurs stepwise, with states backtracked by at most one base pair forming quickly, and further backtracking occurring slowly. Finally, we find that nascent RNA structures act as modulators that either enhance or attenuate pausing, depending on the sequence context.

[1] California Institute for Quantitative Biosciences, QB3, University of California, Berkeley, CA 94720, USA. [2] Department of Physics, University of California, Berkeley, CA 94720, USA. [3] Department of Chemistry and Waksman Institute, Rutgers University, Piscataway, NJ 08854, USA. [4] Department of Biological Chemistry and Molecular Pharmacology, Harvard Medical School, Boston, MA 02115, USA. [5] Department of Chemistry, University of California, Berkeley, CA 94720, USA. [6] Department of Molecular and Cell Biology, and Kavli Energy Nanoscience Institute, University of California, Berkeley, CA 94720, USA. These authors contributed equally: Ronen Gabizon, Antony Lee. Correspondence and requests for materials should be addressed to C.J.B. (email: carlosb@berkeley.edu)

Transcription is a tightly regulated process in which RNA polymerase (RNAP) encodes the genetic information into RNA molecules with either protein-encoding or structural and catalytic roles[1]. After initiating transcription from a promoter, RNAP enters the elongation phase, which consists of periods of processive nucleotide addition interspersed by pauses. Pausing plays critical roles in regulating transcription and in coordinating it with other processes that occur co-transcriptionally, including RNA folding[2], RNA processing and translation[3,4].

The entry into paused states begins with the formation of an elemental paused state with inhibited transcription elongation[5–7]. In *Escherichia coli*, pausing is known to occur at consensus pause sequence elements ($G_{-10}Y_{-1}G_{+1}$, where $-1$ corresponds to the position of the RNA 3′ end and $+1$ corresponds to the next nucleotide to be incorporated)[3,8], through inhibition of the translocation step[8]. The paused states can be further stabilized by the formation of a nascent RNA hairpin[9,10] or by RNAP backtracking[10–12]. However, the events required to enter a paused state from active elongation are not well understood, because those events are beyond the temporal resolution of previous experiments. Optical tweezers experiments[5,13–17] have been used to detect and characterize long-lived pauses (longer than 1 s), but have not been able to reliably and directly detect and characterize short, sub-second pauses. Since the time scale for processive nucleotide addition by RNAP at saturating nucleotide concentrations is ~25–100 ms per nucleotide[18,19], a wide range of physiologically relevant time scales (from ~25 to ~1000 ms) has eluded direct study.

Here, we use a high-resolution optical tweezers assay and developed methods of data analysis to characterize transcription by *E. coli* RNAP through sequence-dependent pause sites with a temporal resolution improved by an order of magnitude (~100 ms). These improvements enable us to answer three key questions about the mechanism of pause entry and the modulation of pauses by backtracking and nascent RNA features. First, we find that sequence-dependent pause sites all involve a slowing of the forward elongation rate of the enzyme RNAP. This result supports a model in which pause sequences reduce on-pathway elongation rates by RNAP, allowing it time to enter off-pathway reactions leading to long-lived paused states. Second, we find that stabilization of sequence-dependent pauses by backtracking involves two consecutive steps: a first step entailing rapid formation of a state that is either non-backtracked or backtracked by at most a single base pair, and a second step entailing slow conversion to a deeper backtracked and longer-lived state. Third, we find that nascent RNA features, such as hairpins, can either increase or decrease the duration of sequence-dependent pauses, depending on the sequence context, most likely through interaction with the pretranslocated state of elongating RNAP.

## Results

### Characterization of pausing at improved temporal resolution.
Previous optical tweezers studies of pausing relied on direct detection of pausing events by identifying time intervals where the measured transcription velocity is below a defined threshold[14,15]. These methods can consistently detect pausing events longer than ~1000 ms, but must rely on extrapolation and/or other assumptions to infer the distributions of events occurring on shorter time scales[20]. The average pause-free velocity of RNAP at saturating concentrations of nucleotide triphosphates (NTP) is 10–40 bp/s, corresponding to a time scale for processive nucleotide addition of ~25–100 ms[14,18,19], an order of magnitude shorter than directly accessible to previous methods (≥ 1000 ms).

To overcome this limitation, we sought to fully characterize the dynamics of sequence-specific pausing, down to the ~100 ms time scale. To this end, we developed a method that can accurately determine (1) the position of RNAP on the template sequence, (2) the time RNAP spends at each pause site for every crossing (the Pause Site Crossing Time), and (3) the pausing efficiencies at each pause site. The method is briefly described below; a detailed description is given in the Supplementary Methods.

The first requirement—accurate determination of the position of RNAP relative to the template sequence—was fulfilled by modifying a procedure from Herbert et al.[13]. We performed transcription experiments on a DNA template (8XHis) containing the T7A1 promoter followed by eight tandem repeats of a 239 bp sequence containing the *his*-leader pause site and four other known sequence-dependent pause sites[5,13] ('a', 'b', 'c', and 'd', Fig. 1a). Single transcription elongation complexes containing biotinylated *E. coli* RNAP halted at position 29 by NTP deprivation were tethered between 1 μm diameter polystyrene beads held in high resolution optical traps (Fig. 1a) Transcription was restarted by moving the bead pair into a region containing saturating NTPs in a laminar flow chamber[21], and subsequent elongation was monitored by measuring the extension of the tether (in nm) over time (Fig. 1b, left). By maintaining constant force with an active feedback loop, we ensured that a constant factor could be used to convert physical distances (in nm) to sequence positions (in base pairs).

We next generated a residence-time histogram (RTH) for each single-molecule trace by sorting the extension data into 0.1 nm bins, and aligned the RTH to estimate, for each trace, the physical length of the 239-bp repeat (in nm). Herbert et al.[13] observed that the aligned RTH yields sharp peaks in defined positions corresponding to the pause sites, and thus estimated the physical repeat length by maximizing the skewness of the aligned residence time values. We used a more general cross-validation approach that assumes only that the RTHs of individual repeats are similar to each other, and identifies for each individual trace the physical repeat length that maximizes the similarity (Supplementary Fig. 1a). The calculated physical repeat lengths varied with force and were well fit with a worm-like chain model (Supplementary Fig.1b). All traces were then aligned to each other and with the known pause site locations using a similar algorithm, in order to calculate the mean RTH at each position.

The resulting sequence-dependent pausing profile is presented in Fig. 1b (right). We detected the strong pause sites characterized by Herbert et al. ('his', 'a', 'b', 'c', and 'd'), nine other sequence-dependent pause sites with shorter residence times (labeled P1-P9), and the almost entirely pause-free regions between the pause sites. The weak pause sites P1-P9 were partially evident in previous studies[13] but with lower resolution. They appear as peaks across all tested conditions and forces, and display the high force sensitivity characteristic of pause sites, in contrast to non-pause sites (as will be shown later, Fig. 2c); these results indicate that P1-P9 are weak pause sites and not random fluctuations in transcription rates. Supplementary Table 1 contains the sequences of the identified pause sites, Supplementary Table 2 contains the average transcription rates in pause-free regions, and Supplementary Table 3 contains the number of traces and repeats aligned for each data set. The 'his', 'a', and 'd' pause sites exhibit three out of three matches to the consensus pause element, $G_{-10}Y_{-1}G_{+1}$[3,8], whereas the 'b' and 'c' pause sites exhibit only two matches. In contrast, the weak pause sites, P1–P9, exhibit at most one out of three matches to the consensus pause element. We also checked if the weak pause sites match the more extensive consensus sequence $G_{-11}G_{-10}T_{-3}G_{-2}Y_{-1}G_{+1}$[3,22], and found only one site (P9) that displayed more than one match. We focused

our study on the five strong pause sites, as well as the site 'P2', which displayed high sensitivity to RNase, as described later.

The second requirement—accurate determination of the pause site crossing times—was fulfilled as follows. We estimated that the location of RNAP at any given position in the data was known to within ± 3 bp (Supplementary Fig. 1f). Given this localization accuracy, we could draw, around each expected pause site, a 6 bp window in which the actual pause site must be located. Since no pause sites were found within 6 bp of one another, each window surrounding a pause site contains ~6 steps, one of which corresponds to the crossing of the pause site itself and the others to the crossing of pause-free sites. Next, we made the key assumption that within each of these 6 bp windows, the position of the pause site corresponds to the slowest step. To estimate the crossing time of the pause site, we denoised the traces using total variation regularization (Fig. 1c, Supplementary Fig. 1e), searched within each 6 bp window for the 1-bp step that took the longest to cross, and took the duration of that slowest 1 bp as the crossing time at that pause site. We measured the crossing time distribution (CTD) at each pause site, as well as in pause-free

regions, for which the data were aggregated into a single distribution (Fig. 1d and Supplementary Fig. 1c). Supplementary Fig. 2e illustrates that the heterogeneity within the reference sites is small compared to the difference between reference sites and pause sites (both P1-P9 and 'a', 'b', 'c', 'd', 'his'), justifying the aggregation of data from all reference sites.

A key feature of the method we use for calculating crossing times is that no element of pause detection is employed. Instead, the method calculates the crossing time for every instance of the tested site, whether a pause was visually apparent or not, yielding full distributions of all crossings of the pause sites. To assess the performance of our method, we simulated transcription traces over 6 bp windows spanning a pause site (see supplementary methods for details), and compared the CTDs obtained from direct pause detection methods[13–15] and CTD's obtained from our method, to the true distribution of crossing times (Supplementary Fig. 1g). We find that at timescales above 250 ms, both methods are relatively accurate, with direct detection slightly underestimating the crossing times and with our method slightly overestimating them (by ~50 ms). However, below 250

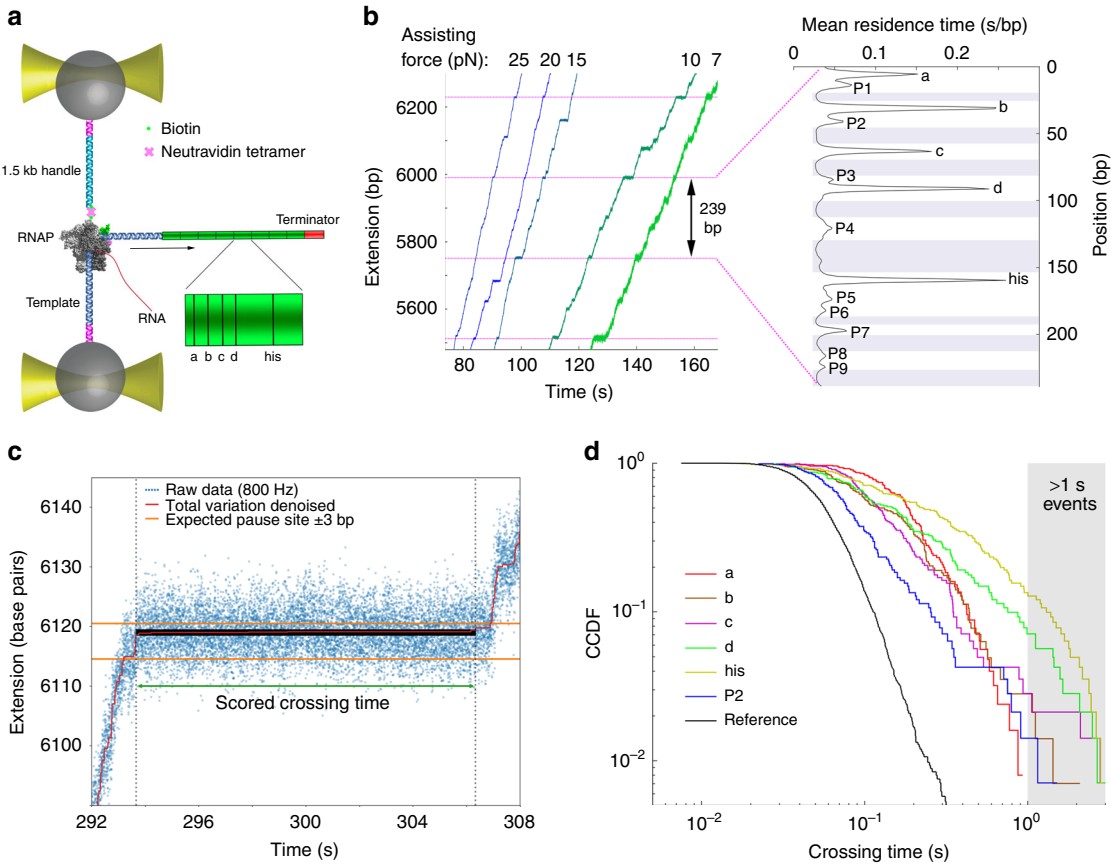

**Fig. 1** Single molecule transcription assay and data analysis. **a** Experimental geometry. Biotinylated *E. coli* RNAP halted on the template DNA was tethered through a neutravidin bridge to a biotinylated 1.5 kb DNA ligated to the oligo-coated bead. The selection of which end of the template DNA to ligate to the other beads enables selection between assisting force (illustrated in panel a) and opposing force geometry. The major pause sites ('his', 'a', 'b', 'c', 'd') are indicated in the sequence of the repeat. **b** (left) representative traces obtained under assisting forces. Dashed magenta lines were added to highlight the locations of the 'his' pause site in the template. The mean residence time histogram shown (right) was calculated by averaging the time spent at each position in the repeat across all traces at all conditions, except RNase data, which was aligned separately (203 traces). Pause sites are marked, and pause-free regions are shaded. For clarity, the mean of measurements up to the 95% percentile is shown to remove to effect of rare pauses occurring outside the pause sites. However, data analysis was performed on the full data set. **c** Total variation denoising and computation of pause site crossing times. The total variation denoising (red) of the raw data (blue) consists of flat segments separated by discrete jumps. One of these segments occurs in the vicinity of an expected pause site. 1 bp windows are drawn in the ±3 bp range surrounding the expected pause site; the window that took the longest to cross (solid black) is used to define the pause site crossing time for the crossing of this pause site. **d** Crossing time distributions for different sites measured at 25 pN assisting force. The complementary cumulative distribution function (fraction of events longer than a given crossing time, CCDF) is plotted. The gray shaded area marks the timescales accessible to previous experiments

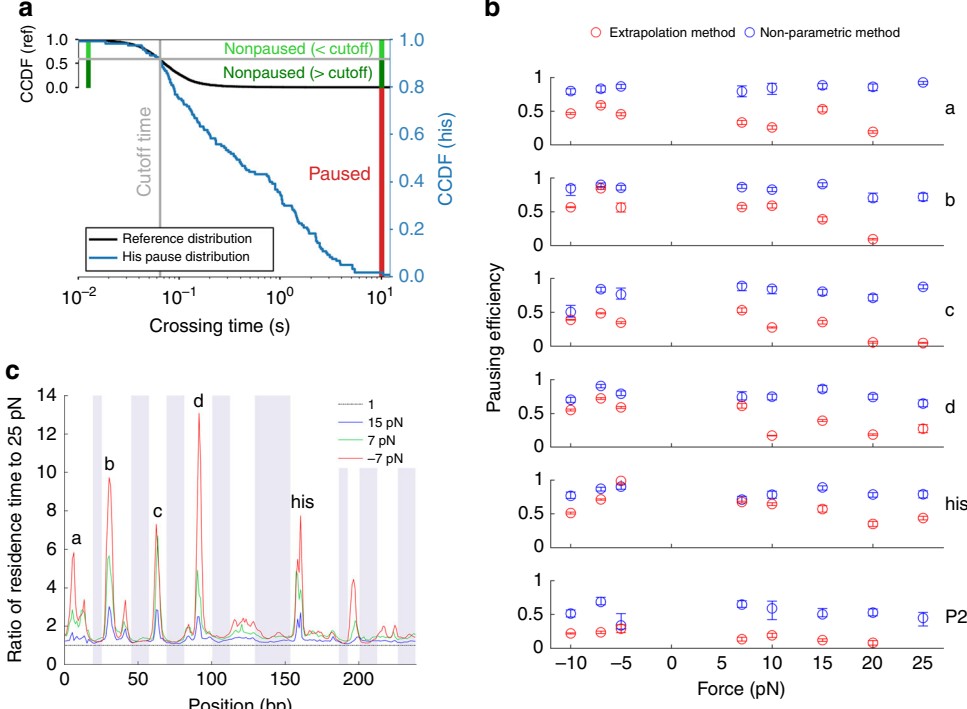

**Fig. 2** Pausing efficiencies are high and force-independent. **a** Description of the method used to calculate pausing efficiencies, illustrated for the 'his' pause at 10 pN assisting force. The reference distribution is rescaled to indicate the overlap between the two distributions below the cutoff time. For the distribution at the 'his' site, all events shorter than the cutoff are classified as non-paused (light green), as well as events longer than the cutoff in the same proportion as in the reference distribution (dark green). The remaining events (red) are classified as paused. **b** Pausing efficiencies at the major pause sites at different forces, calculated using extrapolation of the crossing time distributions above 1 s towards faster times (red), and by our nonparametric method (blue). Error bars indicate 25–75 percentiles for for 100 bootstrapped sets. **c** Residence time histogram ratios. The ratios of the residence times at three forces to the residence times at 25 pN assisting force are plotted. The ratio for 25 pN assisting force, which equals 1 by definition, is plotted as a reference

ms, direct detection fails to detect pauses, leaving 20–50% of the crossings completely uncharacterized. Our method characterizes the entire distribution of pause site crossings, with the said overestimation of their duration. Similar differences are observed for CTDs calculated from the experimental data. The CTDs at pause sites can be clearly distinguished from the distribution measured at non-pause sites under the same conditions using the same method (which we term the Reference CTD), down to a time scale of 100 ms (Fig. 1d).

The third requirement—accurate determination of pausing efficiencies—was fulfilled by employing a non-parametric computational approach, as follows. For each pause site, we define the Pausing Efficiency as the probability that an RNAP molecule will reduce its transcription rate when crossing the site. In previous studies, the number of pauses shorter than 1 s was estimated by extrapolation of the pause lifetime distribution measured at ≥1 s time scales to shorter times. We found that this method significantly underestimates pausing efficiencies (Supplementary Fig. 2b), and fails completely at high assisting forces or at weak pause sites, in which transcription is inhibited but very few ≥1 s events are detected.

In contrast, we have directly measured the full pause site crossing time distributions. Due to the stochastic behavior of single RNAP molecules, at the 50–100 ms time scale, crossing events cannot be unambiguously assigned as pause-free or paused, contrary to very fast (<50 ms) or very slow (>1 s) events, that can be assigned with certainty as pause-free or paused, respectively (Supplementary Fig. 2c). This inherent limitation would remain present even if we could observe single base pair stepping events with infinite temporal resolution. We therefore calculated the pausing efficiency at each site by comparing the

CTD measured there to the reference CTD (measured at non-pause sites). To this end, we found the cutoff time below which the CTD at a given pause site (Fig. 2a, blue curve) is statistically most similar to the reference CTD (Fig. 2a, black curve). The similarity of CTDs below the cutoff at the pause site and at pause free sites suggests that events shorter than the cutoff arise from the same (pause-free) state in both cases (Fig. 2a, light green vertical bar). As stated above, there are also pause-free events longer than the cutoff, which can be estimated from the reference distribution (Fig. 2a, dark green vertical bar). As an example, if 50% of the events in the reference distribution are shorter than the cutoff, then at the pause site there should likewise be an equal number of pause-free events below and above the cutoff. Therefore, the total number of non-paused events at the pause site is thus computed as twice the number of events shorter than the cutoff. The pausing efficiency is calculated as the remaining fraction of crossings, which must arise from the paused state. In fact, this value is likely to be a lower bound on the true pausing efficiency since, contrary to our assumption, even events shorter than the cutoff may arise from the paused state with non-zero probability (Supplementary Fig. 2 and Supplementary Methods). We estimate that true pausing efficiencies may be up to 20% higher than the values we report.

Random pausing events occurring outside the pause sites (termed ubiquitous pauses) were rare compared to previous reports:[14,23] using our current method, the fraction of crossings in non-pause sites longer than 1 s (and therefore detectable as pauses by earlier methods) was <0.03 events per 100 bp for assisting forces, reaching 0.3 events per 100 bp at 10 pN opposing force (Supplementary Fig. 3). We speculate that many pauses assigned as ubiquitous in previous studies with lower spatial

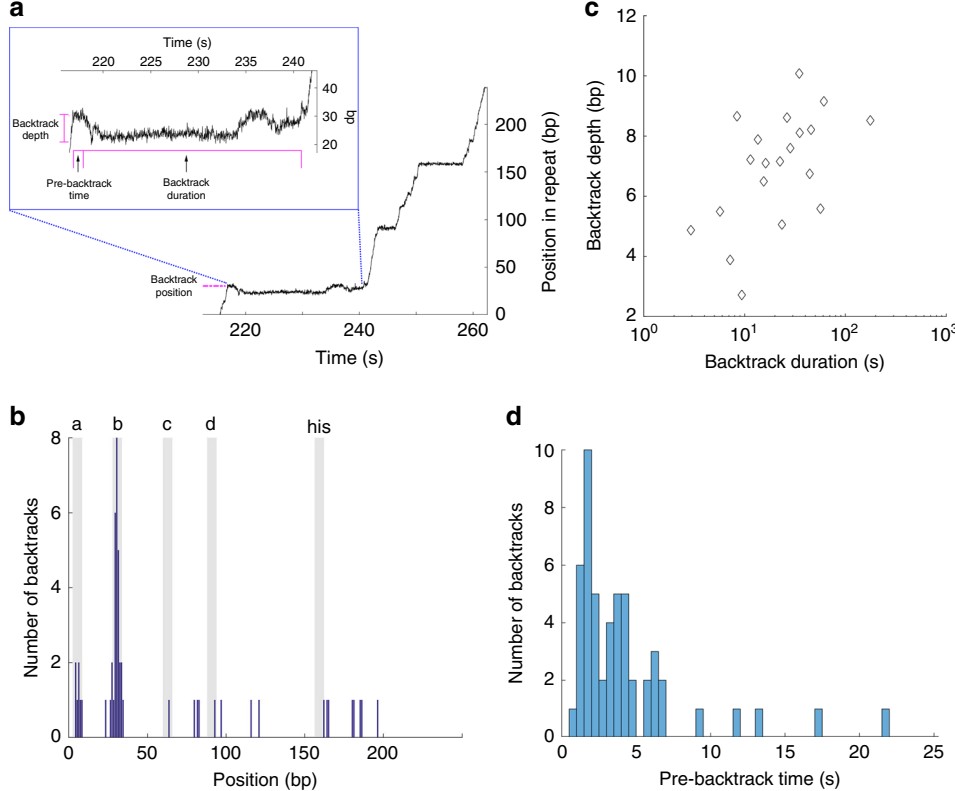

**Fig. 3** Backtracking dynamics. **a** Analysis of backtracking events. The trace shown contains a backtracking event occurring at site 'b'. The backtrack depth, pre-backtrack time and backtrack duration are shown. **b** Histogram of backtracking events by position (blue). Gray zones in the figure indicate the ± 3 bp region surrounding each major pause site. **c** Backtracking depths and times measured at opposing forces for site 'b'. **d** Histogram of pre-backtrack times measured at site 'b' at opposing forces

accuracy were in fact sequence-specific pauses whose exact position in the sequence could not be resolved. In comparison, the fraction of >1 s crossings at pause sites obtained by this same method was typically in the range of 5–35% at assisting forces and 30–50% at opposing forces. Therefore, the assumption that no pausing occurs outside the pause sites would at most cause an additional slight underestimation of the true pausing efficiencies at the pause sites.

**Reduced forward transcription rates at pause sites**. We computed the pausing efficiency for the main sites 'a', 'b', 'c', 'd', and 'his' at different forces using the non-parametric method described above (Fig. 2b, Supplementary Table 4). All pause sites exhibit uniformly high pausing efficiencies (>70–85%) that are independent of force. In other words, almost all RNAP molecules exhibit slower dynamics when crossing sequence-dependent pause sites, regardless of whether they entered an extended paused state. In contrast, the extrapolation-based method consistently underestimates the efficiency, particularly at high forces at sites 'a', 'b', and 'c'. Unlike pausing efficiencies, pause durations are strongly force dependent, as seen both in the distributions of pause site crossing times (Supplementary Figure 3) and in the residence times (Fig. 2c).

**Backtracking occurs in two steps with distinct kinetics**. Using the improved spatiotemporal resolution and positional accuracy of our method, we probed the dynamics of backtracking events down to 2 bp depths under opposing forces; in these conditions, ~1 backtracking event occurred per trace on average (>5 times more frequently than under assisting forces). Specifically, as

illustrated in Fig. 3a, we measured how far RNAP backtracked (backtrack depth), for how long the polymerase paused before it began to backtrack (pre-backtrack time), and how long it spent in the backtracked state (backtrack duration). Backtracking is highly site-specific, with the vast majority of backtracking events occurring at site 'b', and less frequently at site 'a' (Fig. 3b). This site-specificity can be explained by the favorable change in the free energy associated with the backwards propagation of the transcription bubble when RNAP backtracks from site 'b' (see Supplementary Methods and Supplementary Fig. 4a).

We further characterized the backtracking dynamics at site 'b'. First, the backtrack depth and duration were positively correlated (Fig. 3c) with a sub-linear dependence that points to a diffusive backtracking mechanism[15,24]. Return of the RNA 3′-end to the active site of the polymerase does not necessarily imply recovery from the paused state—in many backtracking events, the active site re-aligns with the 3′-end of the transcript only to backtrack again once or several times before actual recovery (Fig. 3a). Second, we found that RNAP does not begin to backtrack immediately upon entering a pause. Instead, it takes at least a second before the enzyme begins to move backwards (with most backtracking events starting 1–10 s after the beginning of the pause, Fig. 3d). This observation indicates that the stabilization of a pause by backtracking occurs in two steps: (1) rapid formation of an initial paused state, which is either non-backtracked or backtracked by 1 bp at most, and (2) slow conversion into a deep and long-lived backtracked state.

To further characterize the backtracking process, we also conducted experiments in the presence of the elongation factor GreB (0.87 μM), which rescues elongation complexes backtracked by as little as 2 bp, but inhibits transcription by non-backtracked

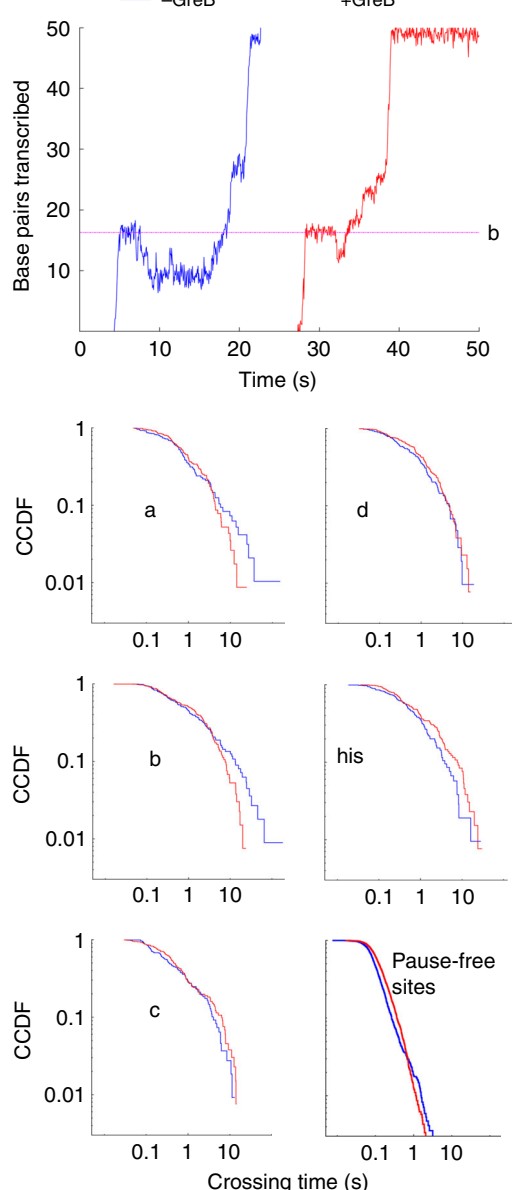

**Fig. 4** Effect on 0.87 μM GreB on transcription dynamics measured at 10 pN opposing force. Top: transcription data in the presence and absence of GreB with backtracking events at site 'b'. Bottom: effect of GreB on the crossing time distributions at the major pause sites. At short time scales, comprising 80–90% of the measured events, GreB slightly increases crossing times. Therefore mean residence times are longer in the presence of GreB at pause sites (Supplementary Fig. 4b). GreB reduces the crossing times at sites 'a' and 'b' for the longest events, indicating that these are backtracked events

RNAP[25] (Fig. 4 and Supplementary Fig. 4b). Transcription data collected in the presence of GreB at 10 pN opposing force, at which backtracking is most favored, display shorter and shallower backtracking events, with rapid recovery indicative of transcript cleavage and transcription directly from the backtracked position, compared to the slower, diffusive return observed in the absence of GreB[11]. GreB has a slight opposite effect on non-backtracked pauses: it slightly increased the crossing times at all time scales at the sites 'c', 'd', and 'his', consistent with the low degree of backtracking observed at those sites (Fig. 4). The effect of GreB was different for sites 'a' and 'b' (Fig. 4)—at short timescales, crossing times at 'a' and 'b' were unaffected, or even slightly

increased in presence of GreB; however, pausing events longer than ~3 s, comprising 20–25% of the events, were highly attenuated. Pause-free sites display similar behavior to pauses 'a' and 'b', but only ~3% of the crossings (corresponding to time scales >0.7 s) are shortened by GreB, indicating that backtracking outside the main pause sites occurs at a very low frequency. This result further confirms that ≥ 2 bp-backtracked states are formed slowly from non-backtracked or 1-bp backtracked paused states.

**Sequence-specific modulation of pausing by the nascent RNA.** We probed the effect of nascent RNA on the transcriptional dynamics by addition of 0.1 mg/ml RNase A[16]. Consistent with reports that pausing is stimulated by the nascent RNA hairpin at the 'his' site, RNase strongly attenuated, but did not abolish the pausing at that site (Fig. 5a). We found that nascent RNA also affects pausing dynamics in other sites, and that the direction and magnitude of the effect vary from sequence to sequence. Pause 'd' was attenuated, though to a smaller extent than 'his', whereas the otherwise weak pause 'P2' was strongly enhanced by RNase. Modulation of pausing by the nascent RNA and backtracking appear to be mutually exclusive, as the backtracking-prone sites 'a' and 'b' did not exhibit sensitivity to RNase, and the sites most sensitive to RNAase do not display backtracking. The residence times at several of the minor pause sites (such as 'P1' and 'P6') were also modulated by removal of the transcript.

Next, we analyzed how the applied force changed the RNase sensitivity of the affected pause sites (Fig. 5b, c). The effect of RNase was consistently stronger at opposing force than at assisting force. This observation could be explained by two scenarios. First, nascent RNA structures may interact more strongly with RNAP in the pre-translocated state, which is favored by opposing force. Second, under opposing force, transcription rates are lower, which may give more time for RNA structures to form, thus enhancing their effect. Simulation of co-transcriptional folding using Kinefold[26] (Supplementary Fig. 5) indicates that under the conditions used in our assay, RNA folding is likely to be fast compared to transcription and therefore the effect of RNA on pausing is unlikely to change due to the small (~15%) variation in transcription rate over the range of forces tested. Accordingly, we tend to favor the hypothesis that nascent RNA structure interacts predominantly with RNAP in pre-translocated state.

Previous studies using bulk transcription assays have found that mutating the nascent RNA hairpin at the 'his' site reduced pause durations with minimal effects on measured pausing efficiencies, while mutations to the consensus pause elements reduced both[27–30]. However, it is unclear whether this conclusion resulted from the limited temporal resolution of the experiments (~10 s) that precluded the detection of short pauses. Using the enhanced resolution in our assay, we tested the effects of RNase on pausing efficiencies at the 'his', 'd', and 'P2' sites. We found that in contrast to its effect on the pause residence times, the effect of RNase on pausing efficiency is minimal, with no significant changes at assisting force and only a ~25% reduction at opposing force for the 'his' site. These observations confirm that the interaction between the nascent RNA and RNAP plays only a minor role in pause entry, and primarily serves to enhance (for 'his' and 'd') or inhibit (for 'P2') the formation of longer lived paused states.

**Discussion**
Sequence-specific pausing involves the formation of elemental paused states[5–7] that are stabilized by processes such as backtracking and RNA hairpin formation. The enhanced temporal resolution of our assay revealed that in addition to these

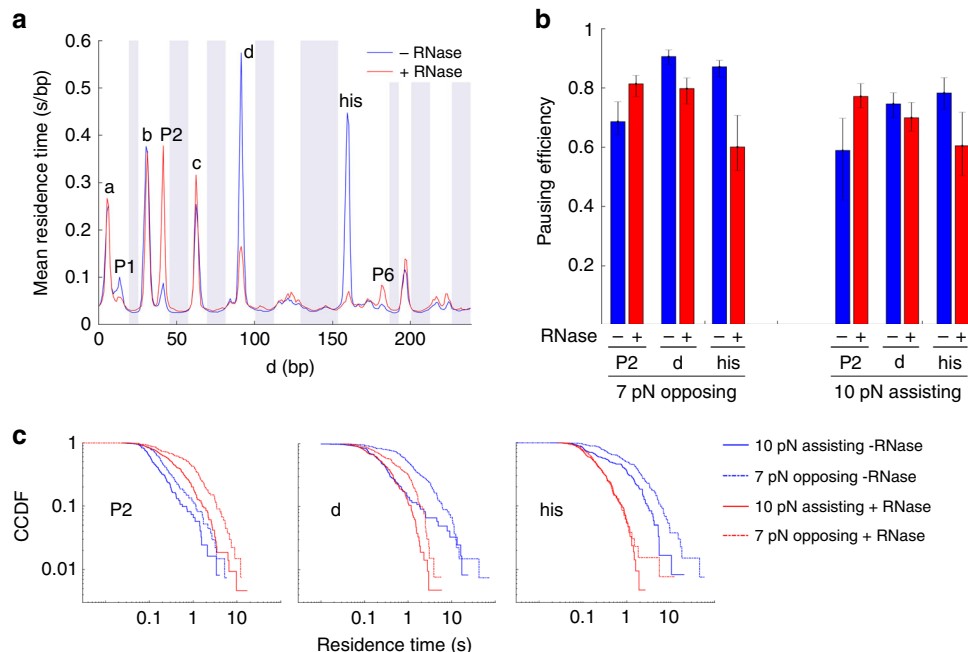

**Fig. 5** Effect of RNase on pausing dynamics. **a** Residence time histograms collected at 7 pN opposing force with and without RNase. **b** Effect of RNase on pausing efficiencies. Error bars indicate 25–75 percentiles for 100 bootstrapped sets. **c** Effect of RNase on residence time distributions at sites 'P2', 'd', and 'his'

mechanisms, pause sequences facilitate pause entry by generally reducing the forward transcription rate of RNAP: nearly all RNAP molecules exhibit slow transcription dynamics when crossing a pause site, even under conditions in which few or no pausing events are long enough to be detected directly using previous methods. This slowing down most likely takes place through sequence-dependent inhibition of forward translocation, as evident from the strong force-dependence of pause durations at these sites. Inhibition of an on-pathway step should result in high and force-sensitive pausing efficiencies, since even at high assisting forces, this step would be slower than the equivalent step at pause-free sites. This inference is further supported by the inhibition of forward translocation of *E. coli* RNAP by consensus pause elements[8] and by studies of *S. cerevisiae* RNA polymerase II, for which sequence-specific translocation barriers that inhibit forward motion of the polymerase have been implicated in pausing[31]. We propose that inhibition of the on-pathway elongation dynamics of RNAP allows time for transitions into stable off-pathway paused states, such as hairpin-stabilized or backtracked-stabilized states[32] (Fig. 6).

The universally high—and force-independent—pausing efficiencies could also be rationalized by the existence of a paused state which is accessed at a very high rate relative to forward transcription, resulting in high pausing efficiencies at all forces. However, having no evidence of this mechanism at this point, we opt for the more phenomenological model of reduction of the on-pathway rate.

Previous studies of transcriptional pausing were blind to the dynamics in the 100–1000 ms time scale[14,15] and estimated the number of pauses shorter than 1 s by extrapolating from the distribution of events longer than 1 s, usually presumed to be exponential. This approach implicitly assumes that the rate of pause escape is the same in both time scales; in fact, it yields a highly inaccurate picture of the pausing dynamics at short timescales (Supplementary Fig. 2b). The Slowest-Crossing method presented here for computing crossing times at sequence-encoded pause sites, enhances the temporal resolution of the dynamics at pause sites, down to ~100 ms. Moreover, the

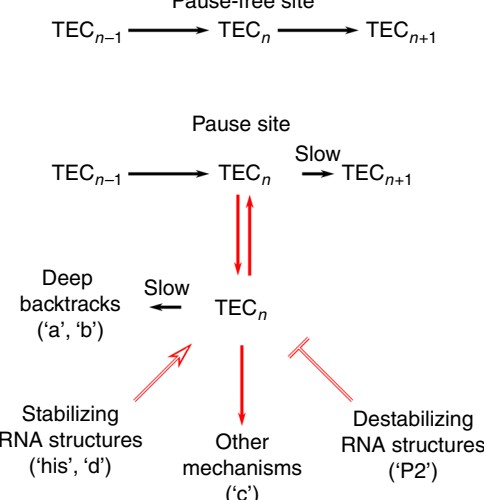

**Fig. 6** Proposed model for transcriptional pausing by *E. coli* RNAP. TEC: Transcription Elongation Complex; the indices $n-1$, $n$, and $n+1$ indicate the length of the RNA product. At pause sites, the paused state is rendered kinetically accessible to the polymerase through a slowing down of the on-pathway forward translocation rate. Depending on the sequence context, this paused state can transition slowly to a $\geq 2$ bp backtracked state (in sites 'a' and 'b'), be stabilized or destabilized by the nascent RNA ('his', 'd', and 'P2') or be stabilized by other mechanisms (such as in site 'c', which exhibited neither backtracking nor RNase sensitivity)

non-parametric method for computing the pausing efficiencies described above relies on the more conservative assumption that pause-free crossings at the pause sites occur via the same mechanism as crossings of non-pause sites and therefore follow the same distribution. Using this approach, we found that pausing efficiencies are much higher than previously determined and are independent of force.

The enhanced temporal resolution of our assay enabled a detailed characterization of the roles played by backtracking and nascent RNA structure in pausing dynamics. Structural studies have found differences between the conformations of 1-bp backtracked and deeper backtracked elongation complexes, both for bacterial and eukaryotic RNAP[33,34]. However, it was unclear whether these structural differences observably modify the dynamics of backtracking. Our results provide direct evidence that the formation of ≥2 bp backtracks involves the rapid and efficient formation of an initial paused state (non-backtracked or 1-bp backtracked) followed by a slower formation of deep backtracked states (Fig. 6). Optical trapping studies on nucleosome-induced pausing have also yielded indirect evidence of similar behavior by yeast RNA polymerase II[35]. Although we could not resolve whether the initial paused state is a non-backtracked or a 1-bp backtracked state, we propose that the two-step nature of backtracking may be a general property of elongation complexes.

As with backtracking, the effects of RNA structure on pausing are site-specific; they vary both in their direction and in their magnitude (Fig. 6). While pausing at the 'his' site is strongly dependent on the hairpin, site 'd' retained significant pausing in the presence of RNase, and the RNA structure primarily stabilizes paused states longer than 1 s. At site 'P2', pausing is inhibited by the nascent RNA. Studies of eukaryotic RNA polymerase II[16] have suggested that RNA structures diminish pausing by generating a physical barrier for RNAP backtracking[36]. However, we observed no significant backtracking at site 'P2', and RNase had no effect on backtracking in general or on the backtrack-prone sites 'a' and 'b'. The CTD at 'P2' was also not affected by GreB, and the effect of RNase was also observed at assisting force, where backtracking is not favored. We therefore conclude that the nascent RNA inhibits pausing at site 'P2' by a distinct interaction with RNAP, and not by inhibiting backtracking. We find that RNA structure primarily affects the duration of the pauses with minimal effects on pausing efficiencies, indicating that entry into the paused state is induced by sequence elements in the template, while the nascent RNA modulates the duration of these pauses, possibly through interaction with RNAP in the pre-translocated state.

Our results contrast with previous reports that detected no effect of nascent RNA folding on pausing[37], and highlight the importance of sequence resolved, high spatiotemporal resolution analysis of pausing. The low temporal resolution and inability to resolve the position of RNAP in previous methods would cause the diverse effects of nascent RNA at various sites to average out. Our study has revealed a far more complex picture of transcriptional regulation by the nascent transcript structure. Since the nascent RNA can bind species such as ribosomes[38] and termination factors[39], it may serve as a fine-tuning element in the transcription cycle, enabling flexible modulation of elongation rates in a context-dependent manner.

Transcriptional pauses play a crucial role in the regulation of gene expression and in the coordination of transcription with other processes. Understanding the molecular transitions that lead from pause-free transcription to paused states requires tools that permit the characterization of pausing dynamics at high spatiotemporal resolution. The development of such tools in this work have resulted in valuable insights into the mechanism of pausing and opens the door to more detailed studies on pause entry of both bacterial and eukaryotic RNA polymerases.

## Methods

**Reagents**. All DNA modifying enzymes were purchased from New England Biolabs. Oligonucleotides were purchased from IDT. Nucleotide triphosphates were purchased from Thermo Scientific, and standard salts and buffer components

were purchased from Sigma Aldrich. Carboxylated 1 μm polystyrene beads were purchased from Bangs Labs. Bacterial strains for protein expression and plasmid propagation were purchased from EMD Millipore.

**Plasmids and DNA templates**. Plasmids pIA1127 (for expression of sigma 70), pIA1234 (for expression of sortagged RNA polymerase), and pIA2–6 (used as a template for preparing DNA handles) were a gift from Irina Artsimovitch (Ohio State University, Department of Microbiology). The (8XHis) template was derived from a plasmid containing the T7A1 promoter, a 29 bp U-less cassette, a ~1 kb downstream spacer region, eight repeats containing the 'his' pause, and finally an *rrnB* T1 terminator sequence[13]. Plasmid for the expression of sortase was a gift from David Liu (Harvard University, Department of Chemistry and Chemical Biology).

**Oligonucleotides used in this study**. For producing 1.5 kb handles from the pIA2-6 plasmid, the following oligonucleotides were used:

For-biotin: 5′ /5Biosg/GAAAGTCCGGCATCTCAATCCC 3′
Rev-BsaI: 5′ ATGATACCGC<u>GAGACC</u>GATGTGGCTTCGGTCCCTTC 3′

In Rev-BsaI the underlined bases denote the BsaI recognition site, which forms a 5′ ACCG overhang after cleavage.

The handle was prepared by PCR and cleaned by standard PCR cleanup, treated with BsaI-HF (5 units per μg DNA for 2 h 37 °C followed by heat inactivation for 20 min at 65 °C) and purified using PCR cleanup.

For modifying bead surfaces, the following oligonucleotides were ordered HPLC purified and used as received:

Bead Amine: 5′ /5AmMC6/TTAATTCATTGCGTTCTGTACACG 3′
Bead CGGT: 5′ /5Phos/CGGTCGTGTACAGAACGCAATGAATT 3′
Bead ACCG: 5′ /5Phos/ACCGCGTGTACAGAACGCAATGAATT 3′

**Preparation of DNA template**. To prepare the DNA template, the 8XHis plasmids was restricted by BsaI-HF (8 units per μg DNA for 2 h at 37 °C) and treated in parallel with shrimp alkaline phosphatase (0.4 units per μg DNA) to generate a linear DNA with distinct, dephosphorylated 5′ overhangs. The enzymes were heat deactivated for 20 min at 65 °C, and the DNA was immediately treated with Klenow 3′−5′ exo- polymerase (1 unit per μg DNA) and 0.1 mM ddATP (to generate an assisting force template) or 0.1 mM ddCTP (to generate an opposing force template). The reaction proceeded for 30 min at 37 °C, followed by heat inactivation for 20 min at 75 °C. The sample was then extracted five times with phenol-chloroform and once with chloroform, ethanol precipitated, and reconstituted in Tris 10 mM pH = 8, 0.1 mM EDTA. The purity of the DNA was assessed to be ~88% by agarose gel electrophoresis.

**Bead coupling to oligonucleotides**. To prepare a double-stranded oligo for coupling, Bead Amine oligo was hybridized to Bead CGGT oligo or to Bead ACCG oligo to generate a double stranded oligo containing a phosphorylated 5′ overhang. Annealing was performed by heating a 1:1 mixture of the oligos in water (0.25 mM each) to 95 °C for 10 min, followed by cooling to room temperature on the bench. This resulted in double stranded oligonucleotides harboring an amine group on one end and either a 5′ CGGT overhang (for assisting force experiments) or a 5′ ACCG overhang (for opposing force experiments) on the other end.

1 μm diameter carboxylated polystyrene beads (Bangs Labs) were coupled to the prepared double-stranded oligos as follows: 10 μl of 10% (W/V) beads were washed 4 times with 200 μl coupling buffer (MES 0.1 M pH = 4.7, 150 mM NaCl, 5% DMSO), and dispersed in 20 μl coupling buffer. All centrifugations took place for 5 min at 4500 g. 10 μl of 20 μM double stranded oligo and 6 μl of 2 M 1-ethyl-3-(3-dimethylaminopropyl)carbodiimide (EDC) were added, followed by vigorous shaking for 2 h at room temperature. At this point another 10 μl of 2 M EDC were added, followed by overnight shaking at room temperature.

The remaining active EDC was then quenched by adding 2.5 μl of 1 M glycine, and the beads were washed 5 times with storage buffer (Tris 20 mM pH = 8, 1 mM EDTA, 0.05% Tween 20, 5 mM sodium azide) with 3 min of sonication between washes. The beads were finally dispersed at a concentration of 1% (W/V) and stored at 4 °C.

**Bead passivation**. The beads were passivated by diluting 6-fold in TE (Tris 20 mM pH = 8, 1 mM EDTA) and addition of β-casein to 1 mg/ml. The beads were vortexed for 10 min, washed once with TE, dispersed at a concentration of 0.2% in TE and stored at 4 °C until the experiment.

**Preparation of sigma 70**. Plasmid pIA1127 was transformed into Rosetta2 bacteria. The bacteria were grown in 2 liters of 2YT medium supplemented with 1% glucose, NPS (25 mM $(NH_4)_2SO_4$, 50 mM $KH_2PO_4$, 50 mM $Na_2HPO_4$), 1 mM magnesium sulfate, 34 μg/ml chloramphenicol and 50 μg/ml kanamycin. The culture was grown at 37 °C to an OD600 of 0.5, transferred to 17 °C and IPTG was added to 0.1 mM. Induction proceeded for 16 h.

For purification, the bacteria were dispersed in 80 ml of buffer A25 (Tris 20 mM pH = 7.5, 0.5 M NaCl, 10% glycerol, 25 mM imidazole, 2 mM beta-mercaptoethanol) supplemented with 0.1 mg/ml lyzozyme and protease inhibitors (Roche). The bacteria

were lyzed by French press, and the lysate was clarified by centrifugation and filtration and loaded on a 5 ml Ni-NTA column. The column was washed with 20 ml buffer A25 and 20 ml A50 (A25 + 50 mM imidazole), and the his-tagged sigma 70 was eluted in A300 (A25 + 300 mM imidazole). TEV protease (expressed in BL21 and purified using Ni-NTA resin followed by size exclusion chromatography[40]) was added at a molar ratio of 1:40, and the reaction proceeded overnight at 4 °C while being dialyzed against A50. The sample was then passed again through an Ni-NTA column. The flowthrough, containing non-histagged sigma 70 was collected, concentrated two-fold, and further purified by gel filtration on a sephacryl S300 column equilibrated with buffer B (Tris 20 mM pH = 7.5, 0.5 M NaCl, 10% glycerol, 1 mM EDTA, 1 mM DTT). The protocol yielded ~ 35 mg of pure sigma 70. Aliquots were flash frozen in liquid nitrogen and stored at −80 °C.

**Preparation of sortagged RNA polymerase holoenzyme**. Plasmid pIA1234 was transformed into Rosetta2 bacteria. Sortag-RNAP was expressed using the same protocol as sigma 70, except that ampicillin was used instead of kanamycin.

For purification we used a modified version of a published protocol[41]. The cells were dispersed in 75 ml of lysis buffer (Tris 50 mM pH = 6.9, 0.5 M NaCl, 5% glycerol) supplemented by 0.1 mg/ml lyzozyme and protease inhibitors, and lyzed by French press. The lysate was centrifuged and filtered, and imidazole was added to 20 mM. The sample was loaded on a 5 ml Ni-NTA column. The column was washed with 30 ml of lysis buffer + 20 mM imidazole and the his-tagged RNAP core enzyme was eluted in lysis buffer + 250 mM imidazole.

To form the holoenzyme, the sample was incubated with a 2-fold excess of purified sigma 70 overnight on ice. The sample was diluted ten-fold with buffer B0 (50 mM Tris pH = 6.9, 5% glycerol, 0.5 mM EDTA, 1 mM DTT) and loaded on a heparin 5 ml column. To avoid overloading the column, the sample was divided into three portions that were loaded separately. A gradient of 50 mM to 1 M NaCl was used to elute the protein. RNAP holoenzyme was separated clearly from excess sigma 70. The sample was dialyzed against buffer B50 (50 mM Tris pH = 6.9, 5% glycerol, 50 mM NaCl, 0.5 mM EDTA, 1 mM DTT), and then purified further on a 1 ml monoQ column using a 50 mM to 1 M NaCl gradient (again, the sample was split into three portions loaded separately). Pure fractions were pooled, dialyzed against storage buffer (20 mM Tris pH = 7.5, 200 mM KCl, 0.2 mM EDTA, 0.2 mM DTT, 5% glycerol), aliquoted, flash frozen and stored at −80 °C.

**Biotinylation of sortag-RNAP**. We obtained a peptide containing an N-terminal GGG tag with a biotin-modified lysine residue (Genscript): GGGGDGDY{Lys (biotin)}. 100 μl of 9.6 μM sortag-RNAP was reacted with a 200-fold excess of biotinylated peptide in 200 μl coupling buffer (Tris 50 mM pH = 7.5, 5 mM CaCl$_2$) containing 2 μM sortase (expressed in BL21 bacteria and purified using Ni-NTA resin followed by size exclusion chromatography[42]). The reaction proceeded for 60 min. At this point, imidazole was added to 25 mM and NaCl to 350 mM, and the sample was passed through 70 μl Ni-NTA beads to remove sortase and unreacted RNAP. The peptide was removed by dialysis into storage buffer, and the biotinylated RNAP was stored in storage buffer at −80 °C.

**Preparation of GreB**. The gene for GreB was cloned into a pET vector by ligation independent cloning (Addgene #29653). The plasmid was transformed in Rosetta 2 cells, the bacteria were grown in 1 liter of 2YT medium supplemented with 1% glucose, NPS (25 mM (NH$_4$)$_2$SO$_4$, 50 mM KH$_2$PO$_4$, 50 mM Na$_2$HPO$_4$), 1 mM magnesium sulfate, 34 μg/ml chloramphenicol and 50 μg/ml kanamycin. The culture was grown at 37 °C to an OD600 of 0.6, IPTG was added to 0.5 mM and transformation proceeded for 4 h at 37 °C. The bacteria were then centrifuged, and dispersed in 40 ml of lysis buffer (Tris 100 mM pH = 7.9, 25 mM imidazole, 1 M NaCl, 2 mM beta-mercaptoethanol) supplemented with 1 mM PMSF and 0.2 mg/ml lyzozyme. The bacteria were lyzed by sonication, and the solution was centrifuged and filtered.

The sample was loaded on a 2 ml Ni-NTA column, washed with 12 ml of lysis buffer, followed by 12 ml of lysis buffer with 50 mM imidazole, and finally eluted with lysis buffer with 300 mM imidazole. TEV protease was added at a 1:10 molar ratio, and the sample was incubated overnight at 4 °C while dialyzing against lysis buffer. The sample was passed again over 1 ml Ni-NTA beads, concentrated to <3 ml and loaded on a sephacryl S100 gel filtration column equilibrated with Tris 25 mM pH = 8, 1 M NaCl, 1 mM EDTA, 1 mM DTT. Fractions containing clean GreB were pooled and concentrated to ~ 50 μM; glycerol was added to 50%; and the protein was flash frozen with liquid nitrogen and stored at −80 °C.

When performing experiments with GreB, the protein was dialyzed first into HEPES 25 mM pH = 8, 1 M KCl, 1 mM DTT and 1 mM EDTA so that it could be mixed into the experimental buffer in precalculated ratios in order to maintain the buffer composition.

All proteins were >95% pure based on SDS–PAGE. Holo-RNAP activity and pausing was confirmed using short template containing a T7A1 promoter, 29-bp U-less cassette and a downstream 'his' site. GreB activity was tested by the rescue of a 2-bp backtracked elongation complex assembled using an RNA oligonucleotide with two mismatched bases at the 3′ end[25].

**Preparation of halted complexes and tether assembly**. Halted complexes were prepared by incubating 2 nM DNA with 10 nM RNAP in TB20 (Tris 20 mM pH = 8, 20 mM NaCl, 20 mM DTT, 10 mM MgCl$_2$, 20 μg/ml casein) for 20 min at 37 °C.

Halted complexes were then ligated to the beads at a ratio of 1 fmol halted complex to 2 μg beads in TB20 in the presence of 0.1 mM ATP and 0.4 units of T4 DNA ligase, for 60 min at room temperature. For 1.5 kb DNA handles, 1 fmol handle was ligated to 3 μg beads.

Following the ligation, heparin was added to 0.4 mg/ml to the beads. To the beads ligated to the DNA handle, a 200-fold excess of neutravidin was added and incubated with the beads for 10 min prior to diluting with experimental buffer. For the halted complex beads, beads were incubated for 10 min with the added heparin before dilution with experimental buffer.

**Instrument design**. Experiments were performed on a time-shared optical tweezers setup modified from the design in Comstock et al.[43,44]. In this configuration, a 1064 nm laser is passed through an acousto-optic deflector, with the laser alternating in position between the two traps every 5 μs. The position of the beads relative to the traps was measured using back focal plane interferometry[45].

**Experimental setup**. The experimental buffer was: HEPES 50 mM pH = 8, 130 mM KCl, 4 mM MgCl$_2$, 0.1 mM DTT, 0.1 mM EDTA, 20 μg/ml heparin and 10 mM NaN$_3$ (added as a singlet oxygen scavenger to reduce the extent of photodamage[46]). NTP concentrations were 1 mM UTP, 1 mM GTP, 0.5 mM ATP and 0.25 mM CTP[18]. When added to the NTP solution, GreB and RNase A concentrations were 0.87 μM and 0.1 mg/ml, respectively. To perform the experiment, we employed a laminar flow setup[21]. The main channel of the chamber was formed by a flow coming from a reservoir containing buffer and a flow from a second reservoir containing the NTP solution. The two flows form well separated regions in the chamber. Beads containing DNA handle were loaded on a side channel connected to the NTP side via a thin capillary, while beads containing the halted complex were loaded on a similar channel connected to the buffer side. Every experiment consisted of the following steps:

1. Trapping a DNA handle beads in the NTP side.
2. Trapping a halted complex bead in the buffer side.
3. Rubbing the beads against each other in the buffer side until a tether is formed (if at all).
4. If a tether is formed and it has the expected length, moving the pair into the NTP side to restart transcription.

Our experiments were performed using an active force clamp that moved one of the traps constantly to maintain the mean force constant. This maintained the force to within 0.1 pN within each trace. However, we discovered that small changes in calibration may arise due to the fact that the calibration and activity measurement were not performed in the same position in the chamber, possibly due to variations in the thickness of the glass or change in refractive index due to the different composition of the buffer in the NTP channel. Therefore, we performed an additional calibration in the NTP channel after the tether broke, and used this to calculate forces. Obviously, the feedback still had to be performed using calibration parameters measured in the buffer channel. This resulted in a small variation in measured force from tether to tether that rarely exceeded 0.5 pN. For each trace, data collection was halted after the tether broke (either prematurely or due to termination) or after very long (>200 s) pauses.

**Data availability**. Data supporting the findings of this manuscript are available from the corresponding author upon reasonable request.

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

## Acknowledgements

This work was supported by: Howard Hughes Medical Institute (C.B., optical trap assembly), NIH Grants R01GM032543 (to C.B., assay development) and R01GM071552 (to C.B., data collection), the US Department of Energy Office of Basic Energy Sciences Nanomachine Program under Contract DE-AC02-05CH11231 (to C.B., computational resources for data analysis), and NIH grant GM041376 to R.H.E. (template and reagent preparation).

## Author contributions

R.G. assembled the instrument, developed data-collection procedures, prepared proteins, collected and analyzed data, and wrote the manuscript. A.L. developed data-analysis procedures, collected and analyzed data, and wrote the manuscript. H.V.M. prepared DNA templates, performed bulk transcription assays, participated in planning, and analysis of single-molecule transcription assays, and provided feedback on the manuscript. R.H.E. supervised bulk transcription assays, participated in planning and analysis of single-molecule transcription assays, and provided feedback on the manuscript. C.B. led the project, analyzed results, and supervised writing of the manuscript.

## Additional information

**Competing interests:** The authors declare no competing interests.

