## [Peer Review File · Nature Communications]

Reviewers' comments:

Reviewer #1 (Remarks to the Author):

In this manuscript, Gabizon and colleagues study the translocation of single RNA polymerases during transcription elongation using optical tweezers. They apply a novel approach for detecting pauses in the RNAP motion, reporting a detection limit ~ 10 times better than in previous work on RNAP. By looking at the properties of the pausing, they study the processes involved in entering into and maintaining paused states. Among their findings is that the polymerase enters into a 1-bp backtracked state before more extensive backtracking occurs, and that structures in the transcript can alter the pausing.

This manuscript presents high-quality data investigating an important topic (the mechanics of transcription elongation), and may in principle be suitable for publication in *Nature Communications*. However, there are a number of concerns to address regarding the robustness of the analysis and interpretation, the strength of the evidence presented, how the results fit into the context of previous work (in particular, how significant the advances reported here actually are), and the clarity of the presentation. I recommend reconsideration after revision to address the points below.

Major points:

1. The new pause-identification analysis method is the core technical advance here that allows shorter-lived pauses to be detected than before. It is therefore essential to demonstrate that the method is reliable and robust, with low rates of false positives and false negatives for identification of pauses and high precision and accuracy for determining pause duration. However, the authors do not seem to have done any such testing of their method (or at least they don't report any such tests), making it difficult to judge how reliable the results reported in the manuscript are. The approach seems reasonable, but a proper evaluation of its performance is needed. I would suggest a simple test applying the entire analysis (from pause identification to characterization) to simulated data with and without pauses of varying durations, where the simulation incorporates noise/drift with the same kind of spectral density as in the measurements. Of course, one expects the performance to be very good for longer pause durations, and to be worse and worse as the duration gets short—the shortest pauses are thus likely to be the least reliable in terms of their identification. Such an analysis will give a better sense of the limits of the method and improve confidence in the analysis of the experimental data.

2. Looking at the data in Figure 1b and comparing to the previous work by Herbert et al using a very similar transcription template (Ref 13), it seems that some of the "newly resolved" peaks (like P2), were present in the earlier data, but were not identified because they are not as well defined as here, which is worth noting. However, some of the peaks newly identified by the authors are not very convincing (especially P3 and P4, but also P5, P6, P8, P9), looking to be of the same size as fluctuations in the data in Ref 13 that were not identified as peaks. Perhaps the earlier work was overly conservative, but in the same token the authors need to convince the reader that the small fluctuations identified as peaks in Fig 1b are not just fluctuations! They're plausible, but not entirely convincing (as opposed to P1 and P2, for example). What is the noise limit for this analysis? The additional testing of the analysis done to answer point 1 will also help here.

3. Information about the number of traces and pauses analyzed should be included in the figures or figure legends.

4. It's known from the work of Adelman et al PNAS (2002) and Neuman et al Cell (2003) that RNAP undergoes ubiquitous pausing, stopping everywhere along the template for times from the sub-second to tens of seconds, comparable to the sequence-dependent pausing. Presumably these

pauses complicate the analysis of the data, by introducing extra pauses that are not located at one of the putative pause sites and generating a background of “unwanted” pauses. For example, the authors state that they make use in the analysis of the fact that no pause sites were located within 6 bp of one another, but such an assumption will no longer hold true when random ubiquitous pausing is added in. A clearer discussion of how ubiquitous pausing is treated and affects the analysis (in particular, showing that the analysis is robust against ubiquitous pauses) is warranted.

5. The pausing that occurs at non-pause sites (used as a reference for comparison to the pause sites) is presumably just ubiquitous pausing. How do the properties of these pauses (durations, rate of pausing) compare to previous work on ubiquitous pausing? Is this distribution of pause times the same in the different regions lacking pause sites? Some extra analysis showing that the “reference” data make sense is needed.

6. A major conclusion from this study is that the forward transcription rate slows when RNAP traverses one of the pausing sites, even if it doesn't pause there. But perhaps this effect is an artifact because of the finite detection window for pauses. Given an exponential (or quasi-exponential) distribution of pause durations, then there are likely to be many short-lived pauses that are not detected because they do not rise above the threshold of ~ 100 ms. These brief pauses would then look like a slower translocation rate around the pause sites in the cases when pauses are not detected. The authors ought to be able to estimate the size of the slow-down one would expect from missing events, to see if this effect could account for their observation of slower motion.

7. The estimation of the pausing efficiency depends very much on how well one can estimate the number of undetected pauses. If missed short pauses contribute to the slower transcription rate in the absence of detected pauses (the “reference” curve), it seems that they should bias the calculation of the pausing efficiency. Have the authors investigated how big this bias might be?

8. One criticism of the manuscript is that, although it seems to promise detailed mechanistic information about transcription elongation, many of the results are primarily descriptive, and their significance remains unclear. The effects of structures in the nascent transcript, for example, are mainly observational, with some mechanistic hypotheses being suggested but not tested. There is also no discussion of how this work relates to previous work studying the effects of structures in the nascent transcript. The most obvious example for comparison is another optical tweezers study, Dalal et al Mol Cell (2006), which found that structures in the transcript had no effect on ubiquitous pausing (contrasting with the results found here for the sequence-dependent pauses). Turning to the backtracking, observing the short-lived 1-bp backtracked state is interesting, but given that it was already suggested by previous work, what insight is gained into backtracking mechanisms from the single-molecule observations? It's unclear how this new work adds to our mechanistic understanding of pausing and backtracking, beyond what was already known. Revisions are needed to clarify what are the advancements in mechanistic understanding, separating what is proven from what is speculation.

9. The authors make a big deal of the “unprecedented” resolution in space and time of their work. This claim needs to be treated more carefully, however. First, the spatial resolution of the measurements does not seem to be noticeably higher than that reported previously in studies of polymerase translocation (such as Ref. 18), so claiming unprecedented spatial resolution does not seem to be justified. It's the improvement in detecting short pauses that is most important in this paper. But this is not actually related to the time resolution of the measurement, rather it is a question of improved sensitivity in pause detection (coming from the new analysis method). The numerous statements touting unprecedented spatiotemporal resolution should therefore be modified to speak of the improved pause detection sensitivity.

10. Overall, the manuscript seems to be aimed at a more technical audience than the general

readership of Nature Communications. I would suggest revision to improve the accessibility to a general audience.

Minor points:

* In the inset of Fig 3a, the polymerase seems to move forward and then back again about 15 s after backtracking. The motion seems to be too large to be drift, what is going on here? I assume that this effect is relatively common, since it's in the trace that was chosen as the representative figure, so some discussion is needed.

* The GreB measurements are nice for showing that a new step in the backtracking process has indeed been resolved. I recommend adding to Fig 4 a panel showing some representative traces with and without GreB.

* Introduction, p. 2: the authors mention that the timescale for adding a nucleotide is ~ 25 – 100 ms, so presumably all times down to ~ 25 ms are physiologically relevant. However, they then go on to imply that timescales below 100 ms are not of interest (where coincidentally ~ 100 ms is the detection limit of their method). This sentence should be reworded to avoid implying something that's not quite right.

* In the SI, the authors reference a figure S12, which doesn't seem to exist.

* Given that there is quite a bit of literature on translocation, pausing, and backtracking during transcription, the reference list seems to be surprisingly short. A more complete discussion of how this work fits into the context of previous work will hopefully lead to a reference list that covers the field more fully.

Reviewer #2 (Remarks to the Author):

Gabizon et al. report an improvement in optical tweezer methodology that enables kinetic analysis of transcription elongation by down to the 100 ms time-scale with near-bp resolution. They couple technological improvements with improvements in computational methods that enable highest temporospatial resolution of single-molecule transcription to date. By applying method to a DNA sequence already highly analyzed by Block group, the authors gain insight into mechanism of pausing that confirms with better technique the conclusions of pioneering study, and also give insight into slow steps in on-pathway transcription that may precede pausing. The work is high quality. The manuscript is well written and will be of interest and accessible to both the single-molecule research community and molecular biologists interested in transcription mechanisms. My critique focuses principally on suggestions for ways authors could improve manuscript in final revision. For expediency, major and minor points are combined in the following list.

1. a minor issue –absence of page or line numbers makes review of this manuscript more difficult.

2. Some may view repetition of the Herbert et al. Cell paper as a lack of novelty, but in terms of scientific strength the opposite is true. By confirming that pauses on the same sequence tested previously arise without backtracking, the authors have advanced general understanding of transcription mechanisms. This is a strength not a weakness of manuscript.

3. page 3 bottom. Authors only consider the most conserved part of pause sequence in matches to weaker pauses. They should consider match to more extensive pause consensus reported in Imashimizu et al., 2015 and Larson et al., 2015.

4. Figure 1D. The distinction in colors for some pauses is hard to see (his pause and P2).
5. On page 5 line 9 "data was aggregated" should be "data were aggregated".
6. page 6, first paragraph. The statements that "strong pauses are...high efficiency" and that "in other words, almost all RNAP molecules exhibit slower dynamics when crossing pause sites" are confusing. The first statement is tautologic and the second statement does not follow from the first (as implied by "in other words"). The authors should rephrase this section to make their intended meaning clear.
7. page 8 "GreB, which rescues elongation complexes backtracked by as little as 2 base pairs, but inhibits transcription by non-backtracked RNAP. Ref 23." Does Ref. 23 report the experimental results establishing this point?. If not, the primary citation should be given.
8. On page 9, first sentence. What concentration of GreB was used for experiments? It should be reported in main text or figure legend.
9. On page 11, last two paragraph. My understanding is that lack of effect of hairpin-RNAP interaction in pause entry and principal effect on stabilizing the pause after entry in the pre-translocated register has been known for some time, which should probably be mentioned (eg., Herbert et al., 2006; Touloukhonov and Landick, 2003 Mol. Cell 12:1125 and earlier references cited therein).
10. First sentence of discussion should give citation for previous description of the elemental pause.
11. Last sentence of first paragraph of Discussion. One of the authors major conclusions is that a slow translocation step provides time for entry into a pause. This conclusion was reached previously by seminal study of Imashimizu et al., 2013 J Mol Biol 425: 697. The authors should cite this precedent and put their findings in appropriate context.
12. Figure 6. It should be made clear that this model is well established for E. coli RNA polymerase but it is less clear that it applies to yeast RNA polymerase II.
13. Bottom of page 13. Authors should make it clear they have not proven 1-bp backtrack in initial pause.

Reviewer #3 (Remarks to the Author):

Gabizon et al employed a well-established single-molecule setup combined with several data analyses innovations to study pausing by bacterial RNA polymerase. The three major findings are: 1) the distribution of the crossing times at pause sites is inconsistent with a simple partitioning between fast un-paused RNAPs and a single population of paused RNAPs identified in the previous single-molecule study of pausing (Herbert et al 2006, Cell); 2) all pause events involving backtracking by several nucleotides are preceded by non-backtracked or 1nt-backtracked pauses; 3) RNA secondary structures such as hairpin have different effects at different pause sites but seem to preferentially act on a pre-translocated RNA polymerase. Overall, the study reports several methodological innovations and interesting observations that further our understanding of transcriptional pausing. The major issue is that Gabizon et al interpret their findings as an evidence for the slowdown of the on-pathway transcription rate at pause sites, while their analyses method seemingly attributes the slowdown to an additional population of brief pauses. Moreover, it is highly uncertain if the scenario involving a slower on-pathway rate can be distinguished from the

alternative scenario involving an additional population of briefly paused RNAP. The analysis methodology employed by Gabizon et al also raises concerns because different analyses routines were employed to compile distributions of crossing times at pause and pause-free sites. These comments and other queries are detailed below.

Major queries:

1. Abstract: "We find that pause sites modify the dynamics of nearly all RNAP molecules, reducing their forward transcription rate, and not, as previously thought, that they only affect the subset of molecules that enter long-lived paused states." Results: "Nearly all RNAP Molecules Exhibit Slow Forward Transcription Rates at Sequence-Dependent Pause Sites".

These statements partially misinterpret the current view on pausing and are quantitatively imprecise. "Nearly all RNAP Molecules" perhaps means "larger fraction of RNAP Molecules than previously thought" and translates into quantitative rather than qualitative difference with the results of the previous study on the topic (Herbert et al 2006, Cell). In reviewer's opinion Gabizon et al provide evidence that strong pause sites contain a substantial fraction of brief pauses in addition to longer-duration pauses. The fraction of briefly paused RNAP roughly corresponds to the difference between the pause efficiencies calculated by non-parametric and extrapolation method presented in Figure 2b. This is a very important result, but it does not change the view on pausing currently prevalent in the field. Instead, Gabizon et al results possibly confirm a long anticipated mechanism postulating that pausing is a multistep process involving initial isomerization into an elemental pause that may get further stabilized into a long-lived pause. Perhaps Gabizon et al results imply that the elemental pause is faster to escape than previously thought. The qualitative difference between Gabizon et al and the previous study is that Herbert et al 2006 reported a single population of paused RNAP whereas Gabizon et al seemingly observe two (or more) populations of paused RNAP with different escape rates due to the improved spatiotemporal resolution.

Another way to look at this issue is to consider that Gabizon et al merely inferred higher pause efficiencies than those estimated by the previously employed extrapolation method in Herbert et al 2006, Cell. However, the re-estimated pause efficiencies do not tell anything about the on-pathway transcription rate as long as they are lower than 100%. Higher pause efficiency means that a larger fraction of transcribing RNA polymerase undergoes isomerization into the off-pathway states. Pause efficiency is the ratio of the rate of isomerization into an off-pathway state and the rate of the on-pathway transcription, so without knowing the former rate little can be said about the latter rate. Next, Gabizon et al state that "We estimate that true pausing efficiencies may be up to 15% higher than the values we report" which may bring the pausing efficiency to 100% at some sites. However, these relatively imprecise estimates are insufficient for proving that the on-pathway transcription rate is reduced. In general, it might be difficult to distinguish if the enrichment in slow crossing times at a sub-second timescale at pause sites is due to brief pauses (e.g. an elemental pause) or due to the slower on-pathway transcription rate. So perhaps both scenarios should be acknowledged and all statements about the slowdown of the on-pathway transcription rate should be softened.

2. The biggest concern about the data analyses is that Gabizon et al picked up the slowest crossing times (out of six sites surrounding a pause site) at pause sites but averaged the crossing times at pause-free sites (6 x 16 sites). It therefore seems that the analysis routine explicitly enriches the distribution at pause sites with the slow crossing times. The issue is lightly discussed in the supplement: "When performing this calculation, we essentially measure the slowest step between six steps taking place in the analyzed window (given our localization accuracy of ± 3 bp). In pause-free regions, this will naturally result in crossing times that are longer than the average pause-free dwell.", but no convincing arguments justifying the comparison of the two distributions compiled in a different way are provided. It feels imperative to additionally analyze the pause-free sites by picking up the slowest crossing times in six-site window and compare the resulting

distribution to that at pause sites. The main conclusions should hold when pause and pause-free sites are analyzed in the same way. The conclusions are arguably not very credible if they are critically dependent on pause and pause-free sites being analyzed differently.

3. Another concern about the data analysis approach is that most pause-free sites are different from each other and possibly have markedly different distributions of crossing times at a sub-second timescale. Accordingly, it is uncertain if comparison of a distribution at a pause-site to an average distribution at pause-free sites is in any way meaningful. Perhaps some playing with the average distribution at pause-free sites is needed to evaluate the robustness of the conclusions. For example, what is the predicted effect of a two-fold variation in the nucleotide addition rate on a distribution of crossing times at pause-free site? In essence, the upper and lower bounds for the pause-free distribution should be delineated and the conclusions should be robust to the variations of the distribution within its upper and lower bounds.

Minor comments

4. "In contrast, we have directly measured pause site crossing times down to 100 ms". This is likely an overstatement. The crossing times were directly measured but it was not exactly known which of them corresponds to a pause crossing. Authors then used several assumptions to infer the distribution of pause crossing times. See also query #2.

5. It would be beneficial to report the results presented in the main text Figure 2b also as a supplementary table.

Response to Reviewers

We would like to thank the reviewers for their important and constructive comments. We have revised the manuscript per their recommendations and respond here to their individual comments:

Reviewer 1:

1. The new pause-identification analysis method is the core technical advance here that allows shorter-lived pauses to be detected than before. It is therefore essential to demonstrate that the method is reliable and robust, with low rates of false positives and false negatives for identification of pauses and high precision and accuracy for determining pause duration. However, the authors do not seem to have done any such testing of their method (or at least they don't report any such tests), making it difficult to judge how reliable the results reported in the manuscript are. The approach seems reasonable, but a proper evaluation of its performance is needed. I would suggest a simple test applying the entire analysis (from pause identification to characterization) to simulated data with and without pauses of varying durations, where the simulation incorporates noise/drift with the same kind of spectral density as in the measurements. Of course, one expects the performance to be very good for longer pause durations, and to be worse and worse as the duration gets short—the shortest pauses are thus likely to be the least reliable in terms of their identification. Such an analysis will give a better sense of the limits of the method and improve confidence in the analysis of the experimental data.

We thank the reviewer for this important suggestion. We performed the simulation proposed by the reviewer, and it revealed important insights providing increased confidence in our method. Briefly, transcription traces across 6 bp windows containing a pause site were simulated by drawing the dwell times at the pause-free sites from an exponential distribution with a mean of 50 ms, and the dwell time at the pause site from an exponential with varying mean values. We added noise directly taken from long-lived pauses observed in experimental data at various forces, in order to fully reproduce the spectral properties of the experiments. The data was analyzed using both direct detection methods and our method. Supplementary figure 1f contains the results for 0.5 second mean pause duration and noise levels corresponding to a tether tension of 10 pN.

We found that for this simple model using exponentially distributed pause durations, both methods perform reasonably well down to pause durations of 250 ms. Pause durations are slightly

underestimated by direct detection and slightly overestimated by our method. However, below 250 ms, direct detection fails completely to detect many pauses, while our method successfully characterizes the entire distribution of pause site crossings, down to the inverse of the pause-free stepping rate, with the same mentioned slight overestimation. Similar trends are observed, for the same time scales, when performing this analysis on experimental data (supplementary figure 1g), increasing our confidence that the simulation adequately assesses the performance of the method. The pause length overestimation is due to some of the crossings of neighboring pause-free sites being included in the 1 bp window used for the calculation; and this effect manifests itself as an overestimation by the duration of approximately one pause-free crossing (~50 ms). We also varied the pausing efficiency in the simulated data sets by drawing a given subset of the dwell times at the pause site from the slow distribution, and the remaining dwell times from the fast (pause-free distribution). We then estimated the measured pausing efficiencies using the method described in the manuscript (supplementary figure S2d). Our results confirmed that pausing efficiencies are underestimated by ~15% using our method. Accordingly, we have included in the text a reference to these simulations and the main conclusions derived from them.

2. Looking at the data in Figure 1b and comparing to the previous work by Herbert et al using a very similar transcription template (Ref 13), it seems that some of the “newly resolved” peaks (like P2), were present in the earlier data, but were not identified because they are not as well defined as here, which is worth noting. However, some of the peaks newly identified by the authors are not very convincing (especially P3 and P4, but also P5, P6, P8, P9), looking to be of the same size as fluctuations in the data in Ref 13 that were not identified as peaks. Perhaps the earlier work was overly conservative, but in the same token the authors need to convince the reader that the small fluctuations identified as peaks in Fig 1b are not just fluctuations! They’re plausible, but not entirely convincing (as opposed to P1 and P2, for example). What is the noise limit for this analysis? The additional testing of the analysis done to answer point 1 will also help here.

We thank the reviewer for raising this point. There are several sources of evidence indicating that these peaks are not simply fluctuations but arise from weak pause sequences:

a. The increased residence times at these sites is observed consistently, across all measured forces and conditions. This consistency can be observed in main figure 5a, supplementary figure 4b, as

well as the additional response figure R1, which includes the residence time histograms for all the conditions we studied. The appearance of the peaks at the same locations under all these conditions strongly suggests that they are not due to random fluctuations.

b. The pause durations at strong pause sites are highly force dependent (figure 2c), whereas dwell times at pause-free sites (i.e., the inverse of the pause-free velocity) only depend weakly on force (figure 2c and table S2). The weak pause sites described here share this force-dependence characteristic with the stronger pauses.

c. Analysis of the crossing time distributions indicates a consistent statistically significant difference between the sites P1-P9 and the reference (non-pause) sites. This difference is large compared to the heterogeneity between non-pause sites, as shown for 10 pN assisting forces in figure R2. This difference is consistent across all tested conditions.

Therefore, we have added a detailed comment incorporating these points in the revised main text.

3. Information about the number of traces and pauses analyzed should be included in the figures or figure legends.

We thank the reviewer for this sensible suggestion. However, as the figures frequently contain data from multiple data sets, in order to avoid excessive figure legend lengths, the number of traces/analyzed repeats for all data sets has been added in supplementary table S3 and referred for the reader in the main text.

4. It's known from the work of Adelman et al PNAS (2002) and Neuman et al Cell (2003) that RNAP undergoes ubiquitous pausing, stopping everywhere along the template for times from the sub-second to tens of seconds, comparable to the sequence-dependent pausing. Presumably these pauses complicate the analysis of the data, by introducing extra pauses that are not located at one of the putative pause sites and generating a background of "unwanted" pauses. For example, the authors state that they make use in the analysis of the fact that no pause sites were located within 6 bp of one another, but such an assumption will no longer hold true when random ubiquitous pausing is added in. A clearer discussion of how ubiquitous pausing is treated and affects the analysis (in particular, showing that the analysis is robust against ubiquitous pauses) is warranted.

The reviewer raises an important point. The works of Adelman et al (PNAS 2002) and Neuman et al (Cell 2003) used lower resolution instruments and did not employ methods to determine the position of RNAP on the template with high accuracy. Later studies, such as the single molecule work by Herbert et al (2006) and genomic approaches such as those used by Larson et al (Science 2014), Vvedenskaya et al (Science 2014) and Imashimizu et al (Genome Biol 2015), indicated that pausing by *E. coli* RNAP is, in fact, highly sequence-specific, and suggested that a large fraction of the detected pauses previously defined as ubiquitous or random were in fact sequence-dependent pauses.

In our data set, the frequency of events longer than 1 second (and therefore robustly detected as pauses by earlier pause detection methods) at the non-pause sites was ~0.03 pause per 100 bp (1 event per 500 windows of 6 bp) at all assisting forces, ~0.1 pause per 100 bp at 5-7 pN opposing force, and 0.3 pause per 100 bp at 10 pN opposing force. These events are, therefore, considerably more uncommon than previously observed (e.g. Neuman et al. (Cell 2003) report 0.5-1.1 pause per 100bp *independently of force*), which again indicates that “ubiquitous” pauses were most likely sequence dependent pauses. In fact, if the ubiquitous pauses described at lower resolution by these authors were indeed not sequence dependent, they should have been more easily detected by our higher spatiotemporal resolution approach. However, having also a highly accurate positioning of the polymerase along the template, we are less likely to score as “ubiquitous”, pauses that are in reality sequence-dependent.

Furthermore, while shorter (undetectable with the previous time resolution, i.e. < 1s) “ubiquitous” pauses may also exist, they would, by definition, occur in a sequence-independent fashion. As such, they would affect (lengthen) both the distributions of crossing times at pause sites and at non-pause sites. This effect would be stronger at non-pause sites: at pause sites, ubiquitous short pauses would often be shorter than the actual, sequence-dependent pause, and thus tend to be masked out. Therefore, the assumption that no pausing occurs outside the pause sites can only lead to **underestimation** of the pausing efficiencies compared to the real values. Again, this effect would be minor due to the scarcity of such events.

5. The pausing that occurs at non-pause sites (used as a reference for comparison to the pause sites) is presumably just ubiquitous pausing. How do the properties of these pauses (durations, rate of pausing) compare to previous work on ubiquitous pausing? Is this distribution of pause

times the same in the different regions lacking pause sites? Some extra analysis showing that the “reference” data make sense is needed.

As discussed in the answer to comment 4, ubiquitous pausing is considerably less common in our data than in previous studies. This is likely due to more accurate assignment of pausing events to specific pause sequences, which was not feasible in earlier studies due to poor accuracy of the location of the polymerase on the template.

The variation between the crossing time distributions at the different reference sites is small compared to the difference from the distributions at pause sites, both strong and weak (figure R2). As a control, we performed pausing efficiency calculations on each reference sites using the other references sites as the background, pause-free distributions. This calculation yielded “pausing efficiency” values centered around zero (supplementary figure 2e, green points), further indicating that references sites can be treated as a single homogenous group to which the distributions at pause sites can be compared.

6. A major conclusion from this study is that the forward transcription rate slows when RNAP traverses one of the pausing sites, even if it doesn't pause there. But perhaps this effect is an artifact because of the finite detection window for pauses. Given an exponential (or quasi-exponential) distribution of pause durations, then there are likely to be many short-lived pauses that are not detected because they do not rise above the threshold of ~100 ms. These brief pauses would then look like a slower translocation rate around the pause sites in the cases when pauses are not detected. The authors ought to be able to estimate the size of the slow-down one would expect from missing events, to see if this effect could account for their observation of slower motion.

The reviewer raises a possible alternative explanation for the slowing down observed when the enzyme traverses all pauses. First, we have now modified the description of the method to clarify that our method does not attempt to **detect** pauses using any predefined criterion. It calculates a crossing time for **every** time RNAP crosses a pause site (or a reference time, for the reference distribution), regardless of whether or not RNAP appeared to have slowed down during that crossing. By definition, short pauses are not missed by this algorithm, since the distributions

contain the durations of **every** crossing of the pause site. As a result, we can measure very short (<50 ms) crossings, especially at non-pause sites. Our claim that we can characterize pausing down to 100 ms time scale means that we can detect clear differences between distributions at pause sites and reference distributions at that time scale, **not** that shorter crossing times cannot be measured. We hope the changes we have made in the description have clarified this point.

We have in fact measured many crossing times at time scales of 50-100 ms. This point can be seen in the distributions shown in figure 1d and supplementary figure 3. However, as explained in the main text and illustrated in supplementary figure 2c, even with infinite spatial and temporal resolution, crossings at the time scales of 50-100 ms cannot be unambiguously assigned as pauses or pause-free events since there would be considerable overlap between the distributions. A situation in which nearly all RNAP molecules crossing a pause site enter a short pause is kinetically indistinguishable from an effective reduction in on-pathway rates. As explained now in the text in addressing the reviewer's point, we have opted for the most parsimonious description of the phenomenon measured.

7. The estimation of the pausing efficiency depends very much on how well one can estimate the number of undetected pauses. If missed short pauses contribute to the slower transcription rate in the absence of detected pauses (the "reference" curve), it seems that they should bias the calculation of the pausing efficiency. Have the authors investigated how big this bias might be?

We thank the reviewer for making a point that required better clarification in the text. As described in our response to comment 6, we measure the crossing time for **every instance** the polymerase crosses the pause site. It is true, however, that at 50-100 ms time scales, individual crossings cannot be unambiguously assigned as pauses, both because of the inherent overlap between the distributions, and because at those time scales, there is a significant probability that one of pause-free sites surrounding the pause site will take longer to cross than the pause site. This effect is by itself a result of the overlap of paused and pause-free distributions, and is observed in simulated data as well (supplementary figure 1f – compare red and blue dashed curves). Therefore, only comparison to distributions obtained from non-pause sites can provide statistically meaningful estimates of pausing efficiencies. From this comparison, we can only obtain, *sensu stricto*, the *additional* amount of pausing that the pause site exhibits relative to non-pause sites. We report this

number as the pausing efficiency (without further qualification), because we explicitly assume that there is no pausing at non-pause sites. If, in fact, some of the crossings at non-pause sites are pauses, this would only shift the actual pausing efficiencies to higher values, thus supporting and strengthening the conclusion of high pausing efficiencies made here.

8. One criticism of the manuscript is that, although it seems to promise detailed mechanistic information about transcription elongation, many of the results are primarily descriptive, and their significance remains unclear. The effects of structures in the nascent transcript, for example, are mainly observational, with some mechanistic hypotheses being suggested but not tested. There is also no discussion of how this work relates to previous work studying the effects of structures in the nascent transcript. The most obvious example for comparison is another optical tweezers study, Dalal et al Mol Cell (2006), which found that structures in the transcript had no effect on ubiquitous pausing (contrasting with the results found here for the sequence-dependent pauses). Turning to the backtracking, observing the short-lived 1-bp backtracked state is interesting, but given that it was already suggested by previous work, what insight is gained into backtracking mechanisms from the single-molecule observations? It's unclear how this new work adds to our mechanistic understanding of pausing and backtracking, beyond what was already known. Revisions are needed to clarify what are the advancements in mechanistic understanding, separating what is proven from what is speculation.

In response to the reviewer's comment, we have now improved the discussion of the effect of RNA structure on pausing. Specifically, since the work of Dalal et al. (2006) was done with lower resolution instrumentation and with no method to measure accurately the position of observed pauses, many of the pauses observed there were likely sequence-dependent pauses, as stated in the response to comment 4. Our use of sequence-resolved data indicates that RNA structure does have a profound effect on pausing dynamics, and that not only the magnitude, but also the sign (enhancement or attenuation) of this effect is sequence specific. It is therefore possible that the work of Dalal et al. did not observe a net effect because the data from pauses enhanced by nascent RNA and pauses inhibited by nascent RNA could not be resolved and, as a result, averaged out.

Regarding the point raised by the reviewer about backtracking, while there is structural

evidence of difference between the structures of 1-bp and >1 bp backtracked complexes, and indirect data from kinetic models of Pol II transcription, our data provides direct evidence and quantitative characterization of the significant kinetic barrier to the entry into deep backtracked states. We believe that this is an important finding that indicates that the structural differences observed between the superficial and deep backtracked states of the enzyme have dynamic consequences on their interconversion, as they are energetically segregated by a significant barrier. In the revised text, we have modified the discussion to clarify and emphasize this point.

9. The authors make a big deal of the “unprecedented” resolution in space and time of their work. This claim needs to be treated more carefully, however. First, the spatial resolution of the measurements does not seem to be noticeably higher than that reported previously in studies of polymerase translocation (such as Ref. 18), so claiming unprecedented spatial resolution does not seem to be justified. It’s the improvement in detecting short pauses that is most important in this paper. But this is not actually related to the time resolution of the measurement, rather it is a question of improved sensitivity in pause detection (coming from the new analysis method). The numerous statements touting unprecedented spatiotemporal resolution should therefore be modified to speak of the improved pause detection sensitivity.

We thank the reviewer for his comment. There exists a fundamental tradeoff between spatial and temporal resolution; for example, the single base-pair resolution data obtained in reference 18 could only be obtained at a non-physiological transcription velocity of ~1bp/s. To our knowledge, our method is the first one that can extract sequence-resolved dynamical information regarding pausing for a polymerase transcribing at physiologically relevant velocities. We maintain that this constitutes a major improvement in the temporal resolution of pause characterization. For the reasons described in the answer to comment 7, we still prefer to avoid the term “pause detection” to describe our method. We have revised the relevant parts of the manuscript to clarify these points.

10. Overall, the manuscript seems to be aimed at a more technical audience than the general readership of Nature Communications. I would suggest revision to improve the accessibility to a general audience.

We thank the reviewer for his comments and constructive criticism. We believe that in addressing his concerns and those of the other reviewers, our manuscript has become significantly clear and more accessible to the broader general readership. Transcriptional pausing is a fundamental feature of control of gene expression and the insights gained here should be of interest to biologists in general and to biophysicists willing to continue to push the envelope on detectability, temporal resolution and accuracy of the phenomenon.

Minor points:

** In the inset of Fig 3a, the polymerase seems to move forward and then back again about 15 s after backtracking. The motion seems to be too large to be drift, what is going on here? I assume that this effect is relatively common, since it's in the trace that was chosen as the representative figure, so some discussion is needed.*

Yes, the reviewer's observation is correct. This occurrence is fairly common and far beyond noise and drift levels. We believe RNAP may diffuse back to align its active site with the 3'-end of the transcript without returning to an active state, and initiate backtracking again. We added a short description of this phenomenon in the manuscript.

** The GreB measurements are nice for showing that a new step in the backtracking process has indeed been resolved. I recommend adding to Fig 4 a panel showing some representative traces with and without GreB.*

We thank the reviewer for this suggestion. We have now added representative traces to figure 4, with a focus on backtracking events in which the effect of GreB is most apparent.

** Introduction, p. 2: the authors mention that the timescale for adding a nucleotide is ~25–100 ms, so presumably all times down to ~25 ms are physiologically relevant. However, they then go on to imply that timescales below 100 ms are not of interest (where coincidentally ~100 ms is the detection limit of their method). This sentence should be reworded to avoid implying something that's not quite right.*

The reviewer's comment is much appreciated. We have now corrected the sentence.

** In the SI, the authors reference a figure S12, which doesn't seem to exist.*

We thank the reviewer for pointing out this omission. The error has been corrected. The figure referred to was meant to be supplementary figure 1f.

** Given that there is quite a bit of literature on translocation, pausing, and backtracking during transcription, the reference list seems to be surprisingly short. A more complete discussion of how this work fits into the context of previous work will hopefully lead to a reference list that covers the field more fully.*

We thank the reviewer for this important comment. We have now deepened our discussion about pausing and extended the reference list accordingly.

Reviewer 2:

1. a minor issue –absence of page or line numbers makes review of this manuscript more difficult.

We thank the reviewer for pointing out this omission. We have now added page numbers to the revised version of the manuscript.

2. Some may view repetition of the Herbert et al. Cell paper as a lack of novelty, but in terms of scientific strength the opposite is true. By confirming that pauses on the same sequence tested previously arise without backtracking, the authors have advanced general understanding of transcription mechanisms. This is a strength not a weakness of manuscript.

We thank the reviewer for the appraisal of our work. It should be noted that Herbert et al. explicitly mentioned not observing backtracking (not to be confused with the hyper/hypo-translocated states discussed in their figure 3); however, their experiments were carried under an

assisting force of 7.3 ± 2.4 pN – a condition under which backtracking was >5 times less frequent than at opposing forces. Backtracking occurred much more frequently at opposing forces and was highly sequence dependent.

3. page 3 bottom. Authors only consider the most conserved part of pause sequence in matches to weaker pauses. They should consider match to more extensive pause consensus reported in Imashimizu et al., 2015 and Larson et al., 2015.

This is an important point. Therefore, we have also tested the weak pause sites against the more extensive consensus site. Even after this analysis, very little match between the weak pause sites and the consensus sequence is observed. It is interesting to speculate that other than to a consensus sequence, these weak pauses may respond to a different (structural, dynamic or energetic) ‘code’ degenerate in the sequence (a sort of *indirect read-out*) and therefore not revealed by it. We have modified the manuscript accordingly.

4. Figure 1D. The distinction in colors for some pauses is hard to see (his pause and P2).

We thank the reviewer for the suggestion. The colors in figure 1D have been altered to improve clarity.

5. On page 5 line 9 “data was aggregated” should be “data were aggregated”.

Yes, this sentence has been corrected in the revised manuscript.

6. page 6, first paragraph. The statements that “strong pauses are...high efficiency” and that “in other words, almost all RNAP molecules exhibit slower dynamics when crossing pause sites” are confusing. The first statement is tautologic and the second statement does not follow from the first (as implied by “in other words”). The authors should rephrase this section to make their intended meaning clear.

We thank the reviewer for making this point. We have rephrased the paragraph to make its

meaning clearer and explicitly state the pause sites discussed ('a', 'b', 'c', 'd', 'his'). Specifically, defining a pause site as "strong" may be confusing and ambiguous, since a pause site may be "strong" in the sense that the average time spent by the polymerase there is high, but still have low pausing efficiency (in other words, a small fraction of polymerases pause there, but those that pause do so for extended periods). However, we argue that the logical relationship between the two parts of the paragraph still holds: since our method of scoring pauses is based on the crossing time of the pause site, it follows that high pausing efficiency implies that almost all RNAP molecules experience slower dynamics when crossing the pause site.

7. page 8 "GreB, which rescues elongation complexes backtracked by as little as 2 base pairs, but inhibits transcription by non-backtracked RNAP. Ref 23." Does Ref. 23 report the experimental results establishing this point?. If not, the primary citation should be given.

We thank the reviewer for pointing this out. There was an error in this reference in the original manuscript. The corrected reference (Tetone 2017) has now been inserted into the revised version.

8. On page 9, first sentence. What concentration of GreB was used for experiments? It should be reported in main text or figure legend.

The concentration of GreB in the experiments was 0.87 μM . We have added this information in the revised manuscript. We thank the reviewer for pointing out this omission.

9. On page 11, last two paragraph. My understanding is that lack of effect of hairpin-RNAP interaction in pause entry and principal effect on stabilizing the pause after entry in the pre-translocated register has been known for some time, which should probably be mentioned (eg., Herbert et al., 2006; Touloukhonov and Landick, 2003 Mol. Cell 12:1125 and earlier references cited therein).

We thank the reviewer for raising this point. We have modified that part of the manuscript and referred to previous studies of pausing at the 'his' site and the contributions of the hairpin to

pausing efficiency and pause duration. Our data confirms that these result hold even at the short time scales characterized in our work.

10. First sentence of discussion should give citation for previous description of the elemental pause.

We thank the reviewer for this suggestion. We have now added the relevant citations at the beginning of the discussion.

11. Last sentence of first paragraph of Discussion. One of the authors major conclusions is that a slow translocation step provides time for entry into a pause. This conclusion was reached previously by seminal study of Imashimizu et al., 2013 J Mol Biol 425: 697. The authors should cite this precedent and put their findings in appropriate context.

We thank the reviewer for this suggestion. We have updated the discussion and added the relevant citations and details.

12. Figure 6. It should be made clear that this model is well established for E. coli RNA polymerase but it is less clear that it applies to yeast RNA polymerase II.

We have updated the figure legend to explicitly state that the model shown only describes transcription by *E. coli* RNA polymerase.

13. Bottom of page 13. Authors should make it clear they have not proven 1-bp backtrack in initial pause.

The relevant sentence has now been modified to clarify this point.

Reviewer 3:

Gabizon et al employed a well-established single-molecule setup combined with several data analyses innovations to study pausing by bacterial RNA polymerase. The three major findings are: 1) the distribution of the crossing times at pause sites is inconsistent with a simple partitioning between fast un-paused RNAPs and a single population of paused RNAPs identified in the previous single-molecule study of pausing (Herbert et al 2006, Cell); 2) all pause events involving backtracking by several nucleotides are preceded by non-backtracked or Int-backtracked pauses; 3) RNA secondary structures such as hairpin have different effects at different pause sites but seem to preferentially act on a pre-translocated RNA polymerase. Overall, the study reports several methodological innovations and interesting observations that further our understanding of transcriptional pausing. The major issue is that Gabizon et al interpret their findings as an evidence for the slowdown of the on-pathway transcription rate at pause sites, while their analyses method seemingly attributes the slowdown to an additional population of brief pauses. Moreover, it is highly uncertain if the scenario involving a slower on-pathway rate can be distinguished from the alternative scenario involving an additional population of briefly paused RNAP. The analysis methodology employed by Gabizon et al also raises concerns because different analyses routines were employed to compile distributions of crossing times at pause and pause-free sites. These comments and other queries are detailed below.

Major queries:

1. Abstract: “We find that pause sites modify the dynamics of nearly all RNAP molecules, reducing their forward transcription rate, and not, as previously thought, that they only affect the subset of molecules that enter long-lived paused states.” Results: “Nearly all RNAP Molecules Exhibit Slow Forward Transcription Rates at Sequence-Dependent Pause Sites”.

These statements partially misinterpret the current view on pausing and are quantitatively imprecise. “Nearly all RNAP Molecules” perhaps means “larger fraction of RNAP Molecules than previously thought” and translates into quantitative rather than qualitative difference with the results of the previous study on the topic (Herbert et al 2006, Cell).

The reviewer raises an important point. However, we believe that the difference between our results and the previous results are significant, both in a qualitative and quantitative manner, for the following reasons:

1. The quantitative difference is large, especially at high assisting forces. The most recent work employing the same template and measuring pausing efficiencies using the extrapolation-based method (Zhou 2011) reported only 55% pausing efficiency at 6 pN opposing force and 25% pausing efficiency at 17 pN assisting force, for the ‘his’ site (for which we consistently measured ~80% or higher at all forces). Such a large quantitative difference has important implications into our understanding of the pausing mechanism at the sites studied, and about the role played by pauses in transcription regulation.
2. In our study, we do not observe any force dependence to the pausing efficiency, in sharp contrast to the results of studies that used the *extrapolation-to-shorter-times* method of pause analysis. This distinct conclusion represents a significant qualitative difference between our study and those of previous authors.

2. In reviewer’s opinion Gabizon et al provide evidence that strong pause sites contain a substantial fraction of brief pauses in addition to longer-duration pauses. The fraction of briefly paused RNAP roughly corresponds to the difference between the pause efficiencies calculated by non-parametric and extrapolation method presented in Figure 2b. This is a very important result, but it does not change the view on pausing currently prevalent in the field. Instead, Gabizon et al results possibly confirm a long anticipated mechanism postulating that pausing is a multistep process involving initial isomerization into an elemental pause that may get further stabilized into a long-lived pause. Perhaps Gabizon et al results imply that the elemental pause is faster to escape than previously thought. The qualitative difference between Gabizon et al and the previous study is that Herbert et al 2006 reported a single population of paused RNAP whereas Gabizon et al seemingly observe two (or more) populations of paused RNAP with different escape rates due to the improved spatiotemporal resolution.

Another way to look at this issue is to consider that Gabizon et al merely inferred higher pause efficiencies than those estimated by the previously employed extrapolation method in Herbert et al 2006, Cell. However, the re-estimated pause efficiencies do not tell anything about the on-

pathway transcription rate as long as they are lower than 100%. Higher pause efficiency means that a larger fraction of transcribing RNA polymerase undergoes isomerization into the off-pathway states. Pause efficiency is the ratio of the rate of isomerization into an off-pathway state and the rate of the on-pathway transcription, so without knowing the former rate little can be said about the latter rate. Next, Gabizon et al state that “We estimate that true pausing efficiencies may be up to 15% higher than the values we report” which may bring the pausing efficiency to 100% at some sites. However, these relatively imprecise estimates are insufficient for proving that the on-pathway transcription rate is reduced. In general, it might be difficult to distinguish if the enrichment in slow crossing times at a sub-second timescale at pause sites is due to brief pauses (e.g. an elemental pause) or due to the slower on-pathway transcription rate. So perhaps both scenarios should be acknowledged and all statements about the slowdown of the on-pathway transcription rate should be softened.

We thank the reviewer for his insightful comment. Indeed, the lack of force sensitivity of the pausing efficiencies, combined with their high values, can indicate two possible mechanisms: in our proposed model, it may reflect a reduced on-pathway transcription rate, resulting in high (and force-insensitive) pausing efficiencies. In the second model (proposed by the reviewer), RNAP enters a short-lived paused state with very high efficiency when entering the pause site. The insensitivity to applied force indicates that the rate of entry into this paused state must be very high relative to forward transcription, making it the dominant pathway even under high assisting forces. In our view, this scenario is kinetically equivalent to a reduction of on-pathway transcription rates. From a structural sense, even the first model (reduction of on-pathway rates) can occur only if the energy landscape for transcription at a pause site differs from the one at non-pause sites, which necessarily indicates that transcription through a pause site involves distinct structural states. Therefore, in the manuscript, we chose to describe the actual phenomenon, measured, i.e., the reduction of transcription which, as the reviewer points out, implies that the polymerase enters alternative states. We have now clarified this point in the text.

Second, since the readout in our method is the crossing time of the pause sites by the polymerase, due to the inherent overlap between the distribution of pause durations and the distribution of pause-free crossing times, it is not surprising that efficiency values of 100% are not (and cannot be) measured. As noted in the reply to the Reviewer 1 point # 6, even a

hypothetical optical tweezers experiment with infinite spatio-temporal resolution would not reach such a value. However, we consider that the high values of the efficiencies, combined with their insensitivity to force, make our suggested mechanism likely. We have now modified the discussion in this respect to clarify this point.

2. The biggest concern about the data analyses is that Gabizon et al picked up the slowest crossing times (out of six sites surrounding a pause site) at pause sites but averaged the crossing times at pause-free sites (6 x 16 sites). It therefore seems that the analysis routine explicitly enriches the distribution at pause sites with the slow crossing times. The issue is lightly discussed in the supplement: “When performing this calculation, we essentially measure the slowest step between six steps taking place in the analyzed window (given our localization accuracy of ± 3 bp). In pause-free regions, this will naturally result in crossing times that are longer than the average pause-free dwell.”, but no convincing arguments justifying the comparison of the two distributions compiled in a different way are provided. It feels imperative to additionally analyze the pause-free sites by picking up the slowest crossing times in six-site window and compare the resulting distribution to that at pause sites. The main conclusions should hold when pause and pause-free sites are analyzed in the same way. The conclusions are arguably not very credible if they are critically dependent on pause and pause-free sites being analyzed differently.

We thank the reviewer for raising this important and crucial point. The wording in the original version of the manuscript was not sufficiently clear. Thus, we have changed the wording in the description of the method to further clarify it. The reference distributions were calculated using *precisely the same method as the pause distributions* (slowest steps in 6-bp windows). The crossings times at the non-pause sites were not averaged, but aggregated (in the sense that all crossing times from all non-pause sites were combined to a single distribution).

3. Another concern about the data analysis approach is that most pause-free sites are different from each other and possibly have markedly different distributions of crossing times at a sub-second timescale. Accordingly, it is uncertain if comparison of a distribution at a pause-site to an average distribution at pause-free sites is in any way meaningful. Perhaps some playing with the average distribution at pause-free sites is needed to evaluate the robustness of the

conclusions. For example, what is the predicted effect of a two-fold variation in the nucleotide addition rate on a distribution of crossing times at pause-free site? In essence, the upper and lower bounds for the pause-free distribution should be delineated and the conclusions should be robust to the variations of the distribution within its upper and lower bounds.

As noted in our response to the reviewer's comment 2, no averaging was performed for the non-pause distributions. However, we have performed several tests to confirm that aggregating the data from non-pause sites into a single distribution is justified:

1. Visual comparison of the distributions reveals that the heterogeneity within the aggregated non-pause sites distribution is small compared to the difference between distributions at non-pause sites and the distributions at pause sites (see attached figure R2). This result is consistent across all measured conditions.
2. We sequentially removed the distributions for individual non-pause sites and calculated the pausing efficiency at those sites, using the other pause-free sites as a reference (supplementary figure 2e). The resulting efficiencies remain peaked strongly around zero, as expected if the individual non-pause site distributions are equivalent to each other.
3. We have performed bootstrapping of the reference distributions at the level of sites (i.e., we compiled reference distributions from different random subsets of the reference sites, and measured the variation such randomization introduces to the pausing efficiencies). This yielded only a small effect on the calculated efficiencies, with a 2% average increase in uncertainties.
4. *"In contrast, we have directly measured pause site crossing times down to 100 ms". This is likely an overstatement. The crossing times were directly measured but it was not exactly known which of them corresponds to a pause crossing. Authors then used several assumptions to infer the distribution of pause crossing times. See also query #2.*

We thank the reviewer for this comment. The section referred to by the reviewer has been clarified, now stating that we have measured the full crossing time distributions and proceeding to describe that crossing times at the 50-100 ms time scale cannot be unambiguously assigned as a pause or pause-free or paused due to the stochastic nature of transcription by RNAP. This leads to the description of our non-parametric method of pause efficiency estimation.

5. It would be beneficial to report the results presented in the main text Figure 2b also as a supplementary table.

We thank the reviewer for his/her suggestion. We have added an additional supplementary table (supplementary table 4) containing pausing efficiency data for the sites 'a', 'b', 'c', 'd', 'his' and 'P2' for all data sets.

REVIEWERS' COMMENTS:

Reviewer #1 (Remarks to the Author):

The authors have provided reasonable answers to the technical questions raised by the referees. The primary remaining issue regards the significance of the results. The revisions to the text have made the significance clearer, but they don't completely dispel my concern that the main results of the manuscript represent more of an incremental improvement on our understanding rather than a major advance (a concern that was also raised by referee 3).

Reviewer #2 (Remarks to the Author):

I have reviewed the authors' revised manuscript as well as the authors' point-by-point response to the reviewers' critiques and find that my concerns have been adequately addressed. I think this version of the manuscript is suitable for publication.

Reviewer #3 (Remarks to the Author):

Gabizon et al made significant improvements to the manuscript and adequately addressed all my queries. I think that the revised manuscript is suitable for publication as it is. If the authors are allowed to make further minor changes to the manuscript, it would be beneficial to clarify why pausing efficiency is independent on force in context of Figure 6. Specifically, it may help readers to better understand the logic of the arguments if the authors indicate which step is dependent on force (presumably the on-pathway step marked "slow") and explain why acceleration/inhibition of that step by assisting/opposing force does not change the pausing efficiency (presumably, the on-pathway slow step remains measurably slower than the pause-free rate even at assisting forces and, consequently, the total fraction of the slower RNAP remains the same).

Response to Reviewers:

Reviewer #1:

The authors have provided reasonable answers to the technical questions raised by the referees. The primary remaining issue regards the significance of the results. The revisions to the text have made the significance clearer, but they don't completely dispel my concern that the main results of the manuscript represent more of an incremental improvement on our understanding rather than a major advance (a concern that was also raised by referee 3).

We maintain that the methodology of pause analysis represents a qualitative break with previous studies. Nonetheless we thank the reviewer for helping us to improve the manuscript.

Reviewer #2:

I have reviewed the authors' revised manuscript as well as the authors' point-by-point response to the reviewers' critiques and find that my concerns have been adequately addressed. I think this version of the manuscript is suitable for publication.

We would like to thank the reviewer again for all the feedback.

Reviewer #3 (Remarks to the Author):

Gabizon et al made significant improvements to the manuscript and adequately addressed all my queries. I think that the revised manuscript is suitable for publication as it is. If the authors are allowed to make further minor changes to the manuscript, it would be beneficial to clarify why pausing efficiency is independent on force in context of Figure 6. Specifically, it may help readers to better understand the logic of the arguments if the authors indicate which step is dependent on force (presumably the on-pathway step marked "slow") and explain why acceleration/inhibition of that step by assisting/opposing force does not change the pausing efficiency (presumably, the on-pathway slow step remains measurably slower than the pause-free rate even at assisting forces and, consequently, the total fraction of the slower RNAP remains the

same).

We thank the reviewer for this suggestion. We have added a clarification of this point in the main text.